# MOOSE-Chem2: Exploring LLM Limits in Fine-Grained Scientific Hypothesis Discovery via Hierarchical Search

**Zonglin Yang**[1,2‡]**, Wanhao Liu**[3,2]**, Ben Gao**[4,2]**, Yujie Liu**[2]**, Wei Li**[5]**, Tong Xie**[6]**,**
**Lidong Bing**[7]**, Wanli Ouyang**[2,8]**, Erik Cambria**[1†]**, Dongzhan Zhou**[2†]

[1] Nanyang Technological University [2] Shanghai Artificial Intelligence Laboratory
[3] University of Science and Technology of China [4] Wuhan University [5] National University of Singapore
[6] University of New South Wales [7] MiroMind [8] The Chinese University of Hong Kong
{zonglin001,cambria}@ntu.edu.sg, zhoudongzhan@pjlab.org.cn

## Abstract

Large language models (LLMs) have shown promise in automating scientific hypothesis generation, yet existing approaches primarily yield coarse-grained hypotheses lacking critical methodological and experimental details. We introduce and formally define the new task of *fine-grained scientific hypothesis discovery*, which entails generating detailed, experimentally actionable hypotheses from coarse initial research directions. We frame this as a combinatorial optimization problem and investigate the upper limits of LLMs' capacity to solve it when maximally leveraged. Specifically, we explore four foundational questions: (1) how to best harness an LLM's internal heuristics to formulate the fine-grained hypothesis it itself would judge as the most promising among all the possible hypotheses it might generate, based on its own internal scoring-thus defining a latent reward landscape over the hypothesis space; (2) whether such LLM-judged better hypotheses exhibit stronger alignment with ground-truth hypotheses; (3) whether shaping the reward landscape using an ensemble of diverse LLMs of similar capacity yields better outcomes than defining it with repeated instances of the strongest LLM among them; and (4) whether an ensemble of identical LLMs provides a more reliable reward landscape than a single LLM. To address these questions, we propose a hierarchical search method that incrementally proposes and integrates details into the hypothesis, progressing from general concepts to specific experimental configurations. We show that this hierarchical process smooths the reward landscape and enables more effective optimization. Empirical evaluations on a new benchmark of expert-annotated fine-grained hypotheses from recent literature show that our method consistently outperforms strong baselines.[1]

## 1 Introduction

Large language models (LLMs) have increasingly been applied to assist scientific research (Luo et al., 2025), with one of the most ambitious applications being the automated discovery of novel and valid scientific hypotheses. However, current methods produce hypotheses that are criticized for being overly coarse, lacking sufficient detail, offering simplistic suggestions, or omitting concrete implementation strategies (Wang et al., 2024; Hu et al., 2024; Si et al., 2025).

---

[1]All code and data can be found in `https://github.com/ZonglinY/MOOSE-Chem2`
[‡]Contribution during internship at Shanghai Artificial Intelligence Laboratory. [†]Corresponding author.

39th Conference on Neural Information Processing Systems (NeurIPS 2025).

We present the first systematic investigation into how LLMs can be leveraged to formulate fine-grained scientific hypotheses—those enriched not only with major concepts but also with precise methodological details and clearly specified experimental configurations. For example, a coarse-grained hypothesis in chemistry might state, "*synthesize hierarchical 3D copper*," while a fine-grained counterpart could elaborate, "*Copper foils are chemically oxidized by immersion in a solution of 0.5 M ammonium persulfate and 2 M sodium hydroxide for 15 minutes at room temperature, forming a pentagonal hierarchical* $CuO$ *nanostructure.*" Such fine-grained hypotheses significantly enhance clarity, feasibility, and experimental implementability.

Formally, we define the task as generating a fine-grained hypothesis given a research background—comprising a research question and a survey of established methodologies—and a coarse-grained hypothesis direction. We show that fine-grained scientific hypothesis discovery is a combinatorial search problem, as it requires selecting and composing a coherent set of concrete details from a vast space of plausible options—making it particularly challenging in practice. The difficulty is compounded by the fact that scientific hypothesis discovery is an inherently out-of-domain (OOD) problem: the correctness of a hypothesis is fundamentally *unknown* at the time of formulation.

In this work, we focus on the pre-experimental stage of discovery, mirroring how human scientists—prior to empirical testing—iteratively search through the hypothesis space using heuristics and domain knowledge to identify the hypothesis they themselves would judge as the most promising among all plausible candidates they could think of during the hypothesis search process.

Our goal is to emulate this cognitive search process using LLMs, which increasingly rival human scientists in heuristic reasoning and scientific knowledge understanding. This motivates our central research question (*Q1*): *how to best harness an LLM's internal heuristics to formulate the fine-grained hypothesis it itself would judge as the most promising among all possible hypotheses it might generate?* We conceptualize a hypothesis space where each point along the input dimensions (the $x$-axis, potentially multidimensional) represents a candidate hypothesis, and each point is assigned a reward value (on the $y$-axis) by the LLM based on its internal heuristics. This defines a reward landscape over the hypothesis space, with the highest peak corresponding to the hypothesis the LLM internally judges as most promising. Framed this way, *Q1* becomes an optimization problem: *how can we navigate this landscape to find stronger local optima—or ideally the global optimum—thus eliciting the best fine-grained hypothesis the LLM can generate?*

A straightforward baseline is greedy search over the reward landscape. However, its non-convex and complex structure makes naive greedy strategies prone to poor local optima. To address this, we propose a hierarchical search framework that explicitly models how a finite-capacity reasoning agent—human or LLM—navigates the hypothesis space. Specifically, it first explores higher-level conceptual spaces and then incrementally refines into more specific detail spaces. This hierarchical approach smooths the reward landscape at each hierarchy level—especially at higher, more abstract levels—enabling convergence to superior local optima compared to greedy search and greedy search with self-consistency (Wang et al., 2023). The proposed framework naturally scales with the capability of the underlying LLM's heuristics, yielding better optima as those heuristics become stronger.

Having investigated how to identify stronger local optima in *Q1*, we now turn to our second question (*Q2*): *whether hypotheses judged better by LLMs exhibit stronger alignment with ground-truth hypotheses?* To rigorously address *Q2* while avoiding data contamination, we construct a benchmark of research backgrounds paired with expert-annotated fine-grained hypotheses from chemistry papers published after January 2024, ensuring these examples were unseen by our LLM (GPT-4o-mini, October 2023 cutoff). Using this benchmark, we indirectly evaluate *Q2* by comparing the recall of hypotheses discovered by our hierarchical approach—which locates better LLM-internal local optima—with hypotheses identified by baseline methods. Our results consistently show that hypotheses generated by our method achieve higher recall than those from baselines, providing empirical support for the reliability of the LLM's internal reward signal in guiding fine-grained hypothesis discovery.

Until now, the reward landscape guiding hypothesis search has been defined by a single LLM serving as the evaluator. We now address our third question (*Q3*): *whether defining this landscape with an ensemble of diverse LLMs of similar capacity yields better outcomes than using equally sized ensembles of the strongest LLM within that group.* Our experiments show that ensembles of repeated instances of the strongest LLM consistently outperform equally sized ensembles of diverse models, suggesting that peak model quality is more critical than model diversity in this setting.

Finally, we consider a fourth question (*Q4*): *whether an ensemble of identical LLMs provides better reward landscape than a single instance of the same LLM*. While *Q3* compares ensembles of different models, *Q4* isolates the effect of aggregation alone by controlling for model identity. We find that even identical LLMs, when sampled independently and aggregated via summarization, yield a reward signal that better captures novelty without sacrificing overall quality—highlighting a subtle but important dimension in optimizing hypothesis discovery.

Notably, while our experiments focus on chemistry, the task formulation, methodology, and analysis of *Q1-Q4* are discipline-agnostic. The only domain-specific component is the manually designed hierarchy (used by the methodology), which can be instantiated for each new discipline encountered.

Overall, the contributions of this work are:

1. We introduce and formalize the *fine-grained scientific hypothesis discovery* task as a combinatorial optimization problem, and release a post-2024 benchmark with expert-annotated fine-grained hypotheses, specifically designed to prevent data contamination.

2. We explore the limits of LLMs for fine-grained scientific hypothesis discovery by framing it as an optimization problem over a reward landscape defined by LLM heuristics, with pairwise comparisons serving as the gradient signal. We also propose a hierarchical heuristic search framework that theoretically smooths the reward landscape, reduces search complexity, and identifies superior local optimum by interpolating among discovered optima. Empirically, this framework consistently outperforms strong baselines in locating better local optima.

3. We analyze this optimization formalization through 4 foundational research questions.

## 2 Methodology

### 2.1 Background and Task Motivation

Yang et al. (2025) assume that many chemistry hypotheses can be constructed from a research background $b$—typically including the research question and/or background survey—and a set of inspirations $i_1, \ldots, i_k$, representing concepts or findings from the literature. It can be formulated as:

$$h = f(b, i_1, \ldots, i_k) \tag{1}$$

In practice, however, most hypotheses $h$ generated from Equation 1 tend to be coarse-grained: while they form cohesive associations between $b$ and the $i$, they often lack clear hypothesis specification and the detailed experimental configurations required for direct implementation in a laboratory setting. Additionally, many such hypotheses contain redundant elements—either due to the inclusion of unnecessary inspirations or from noise present in the literature that is unrelated to the core knowledge intended for hypothesis construction.

### 2.2 Problem Formulation: Fine-Grained Hypothesis Generation as Combinatorial Search

Let $h_c$ be a coarse-grained hypothesis direction and $h_f$ its fine-grained counterpart, defined as:

$$h_f = \{h_c, d_1, \ldots, d_m\} \tag{2}$$

Here, $\{h_c, d_1, \ldots, d_m\}$ denotes the meaningful integration of edits $d_1, \ldots, d_m$ into $h_c$, resulting in a coherent, fine-grained hypothesis. Each edit $d$ corresponds to either (1) the addition of a fine-grained detail to an existing concept $i$ or the introduction of a new concept $i$ into $h_c$, or (2) the deletion of a redundant detail or concept from $h_c$. We define two edit candidate sets: $D^+$, containing all details and concepts that may be added to $h_c$; and $D^-$, containing all details and concepts within $h_c$ that may be removed. The overall edit space is then given by $D = D^+ \cup D^-$.

Inspired by coarse-to-fine strategies in computer vision—where a coarse image is first generated and then refined with fine-grained details (Tian et al., 2024)—we formulate the transition from $h_c$ to $h_f$ as an additional step building on Equation 1, which provides the initial $h_c$.

$$P(h_f|b, h_c) = P(\{d_1, \ldots, d_m\}|b, h_c, D) \tag{3}$$

This formulation turns $P(h_f \mid b, h_c)$ into a combinatorial optimization problem, where the objective is to select a subset of edits $d_1, \ldots, d_m \subseteq D$. Let $|D| = n$ and $|d_1, \ldots, d_m| = m$. The search space has at least combinatorial complexity $C_n^m = \frac{n!}{m!(n-m)!}$. This makes the problem particularly challenging due to three factors: (1) both $m$ and $n$ are unknown; (2) the candidate set $D$ is itself implicit and potentially very large; and (3) the edits $d_i$ are not independent—errors early in the reasoning chain can propagate and impair later decisions.

## 2.3 Algorithmic Motivation for Hierarchical Heuristic Search (*HHS*)

Fine-grained hypothesis generation is generally intractable due to the exponential growth of the search space, where the candidate set $|D|$ is often large or prohibitively so.

A notable exception occurs when the problem exhibits an optimal substructure—i.e., an optimal solution can be composed from the optimal solutions to its subproblems. This principle underlies dynamic programming, where solutions are built incrementally from smaller subproblems (first try to obtain the optimal solution for a smaller subproblem and then iteratively find the optimal solution for larger subproblems).

We observe that fine-grained hypothesis generation exhibits an optimal substructure. Specifically, the edits $d_1, \ldots, d_m$ can be organized hierarchically: some address high-level concepts (e.g., functional groups, catalyst classes), while others specify low-level details (e.g., reagents, catalysts, temperature, concentration). We assume these edits can be partitioned into $p$ hierarchical levels ($p > 1$), with higher levels corresponding to finer details. Then the overall problem can be seen as to determine $d$ in $\{1, \ldots, p\}$ hierarchies. The subproblem of it can be seen as the determination of $d$ in $\{1, \ldots, p-1\}$ hierarchies, etc. Then it is obvious that the optimal solution of a problem can be derived from the optimal solution of its subproblem, etc.

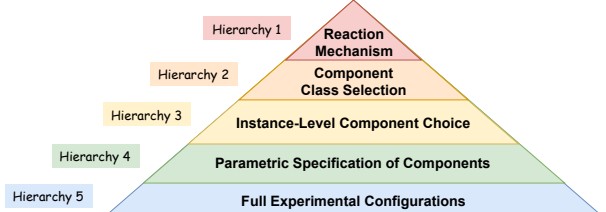

Figure 1: Hierarchies designed for chemistry and material science by PhD-level domain expert.

Figure 1 illustrates an example hierarchical decomposition for chemistry, developed in collaboration with domain experts (PhD-level chemists). The hierarchy spans from high-level mechanistic intent to low-level experimental configurations, reflecting the granularity typically considered when translating a conceptual hypothesis into a testable laboratory procedure in chemistry.

Now we have simplified the problem of determining $d$ in all $p$ hierarchies into the iteration of determining $d$ in each hierarchy sequentially. Nonetheless, even within a single hierarchy, the number of candidates remains combinatorially large.

A practical approach to this combinatorial complexity is to use heuristics that approximate solutions rather than exhaustively searching for exact ones. This aligns with how chemists refine hypotheses: given that $h_c$ often represents an unexplored direction, while individual details may be retrievable from existing databases, the complete set of details is rarely available. Instead, chemists often rely on domain knowledge and intuition to heuristically identify and progressively integrate plausible details.

Analogously, we propose to leverage LLMs' internal heuristics to guide the search for $d$ at each hierarchical level. As LLMs advance, their heuristics—emerging from pretraining over extensive scientific corpora—will increasingly approximate, and may surpass, those of human experts. The proposed framework naturally scales with the strength of these heuristics, yielding increasingly better optima for fine-grained hypothesis discovery as LLM capabilities continue to grow.

In this setting, the candidate space $D$ is not explicitly enumerated but is implicitly embedded within the LLM's internal knowledge and reasoning capabilities. The LLM does not select $d$ from a predefined list, but rather proposes candidates by navigating this latent, heuristic-driven space.

## 2.4 Hierarchical Factorization of the Search Problem

For formalization, we partition the implicit candidate space $D$ into $p$ hierarchical levels, where $D^{(i)} \subseteq D$ represents all potential edits at level $i$, and $D^{*(i)} \subseteq D^{(i)}$ denotes the (unknown) ground-truth edits. The $j$-th ground-truth edit at level $i$ is denoted as $d_j^{*(i)} \in D^{*(i)}$. Since $D$ is implicitly determined by $h_c$, we have $P(D \mid h_c) = 1$, and explicitly condition on $D$ for clarity in the subsequent factorization. Applying the chain rule hierarchically, Equation 3 can be reformulated as:

$$P(h_f \mid b, h_c) = P\left(\{D^{*(1)}, \ldots, D^{*(p)}\} \mid b, h_c, D\right) \tag{4}$$

$$= \prod_{i=1}^{p} P\left(D^{*(i)} \mid b, h_c, D^{*(<i)}, D^{(i)}\right) \tag{5}$$

$$= \prod_{i=1}^{p} \prod_{j=1}^{|D^{*(i)}|} P\left(d_j^{*(i)} \mid b, h_c, D^{*(<i)}, d_{<j}^{*(i)}, D^{(i)}\right), \tag{6}$$

where $D^{*(<i)} = \{D^{*(1)}, \ldots, D^{*(i-1)}\}$ and $d_{<j}^{*(i)} = \{d_1^{*(i)}, \ldots, d_{j-1}^{*(i)}\}$.

The key advantage of this hierarchical factorization is that at each level $i$, the search is restricted to the reduced candidate set $D^{(i)}$ rather than the full space $D$, significantly narrowing the search space. Moreover, as we will show in § 2.6, this hierarchical decomposition smooths the reward landscape at each hierarchy level, facilitating more stable optimization and enabling the discovery of stronger local optima in the hypothesis space.

## 2.5 LLM-Based Implementation of Hierarchical Heuristic Search

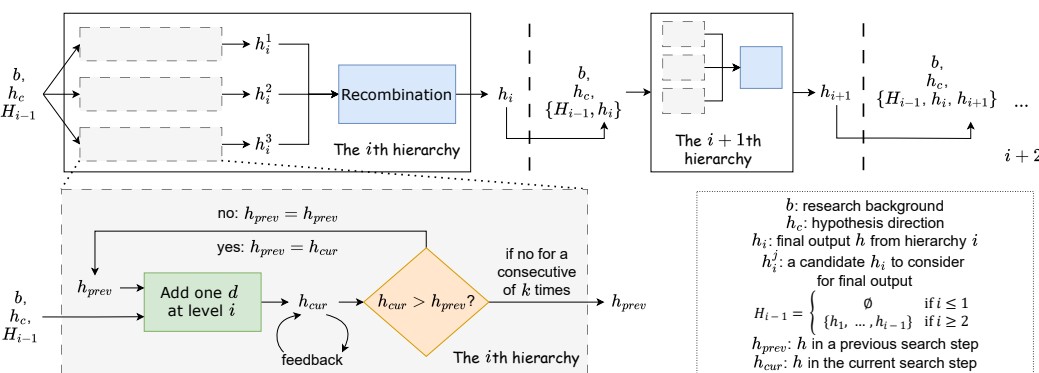

Figure 2: Overview of the proposed Hierarchical Heuristic Search (HHS) framework.

We implement *HHS* as an LLM-driven agentic process that directly follows the hierarchical factorization formalized in Equation 6. As shown in Figure 2, at each hierarchy level $i$, $H_{i-1}$ represents the accumulated edits from all previous levels, corresponding to $D^{*(<i)}$. Within the current level, $h_{prev}$ denotes the partial hypothesis incorporating the edits selected up to step $j-1$, i.e., $d_{<j}^{*(i)}$. The candidate set $D^{(i)}$ is not explicitly enumerated but emerges implicitly from the LLM's internal heuristics, conditioned on the background $b$, the hypothesis direction $h_c$, and the edits selected so far.

Specifically, the search for a local optimum $h_i^j$ begins from the initial point $h_{i-1}$, using contextual information from $b$, $h_c$, and $H_{i-2}$. For hierarchy level $i = 1$, we set $h_0 = h_c$ and $H_0 = \emptyset$, making the hypothesis direction $h_c$ the starting point.

At each iteration, the "*Add one d at level i*" module prompts an LLM to propose an edit $d$ to $h_{prev}$, producing a candidate $h_{cur}$, which is then refined once for validity, novelty, and specificity. The "$h_{cur} > h_{prev}$" module evaluates whether the new hypothesis improves upon the previous one via LLM-based pairwise comparison, serving as an *internal gradient signal* for hypothesis optimization.

This search process continues until no further improvement is observed over $k$ consecutive steps (default $k = 3$), at which point the current hypothesis is accepted as a local optimum. Each edit $d$ may involve either an addition or a deletion, allowing the search path to include retrospection and self-correction as needed.

Within each hierarchy level, we adapt the design of an evolutionary unit (Yang et al., 2025) to our task, where the search for the local optimum $h_i$ is independently repeated multiple times (set to three in our implementation), yielding several local optima $h_i^1$, $h_i^2$, and $h_i^3$. These candidates are then passed to a *recombination* module, which integrates their complementary strengths to interpolate a potentially superior local optimum $h_i$ within the subspace spanned by $h_i^1, h_i^2, h_i^3$.

## 2.6 Theoretical Analysis: Smoothing Effects of Hierarchical Heuristic Search

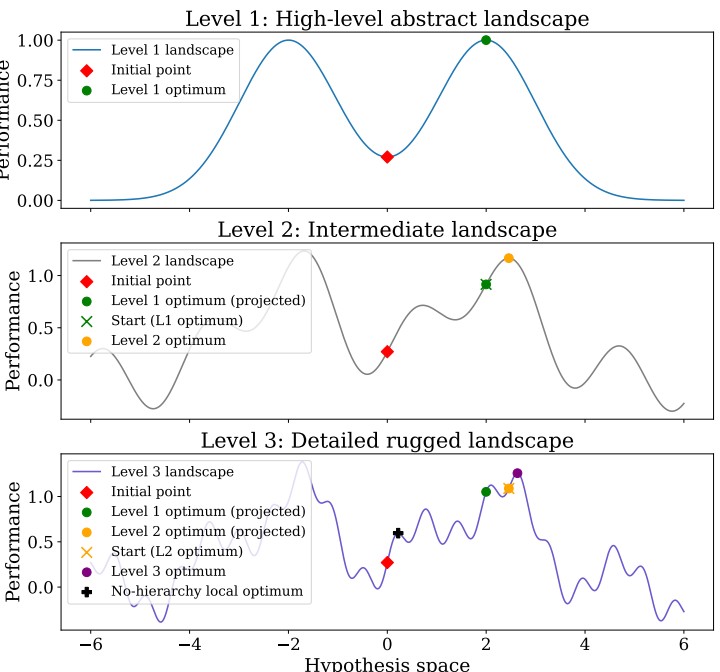

Figure 3: The smoothing effect of hierarchy on the reward landscape of hypothesis space.

A key observation is that a hypothesis candidate's performance at lower (more abstract) hierarchy level can be viewed as an aggregated estimate—approximating an average or soft maximum—of its higher-level subspace (more concrete). For instance, when evaluating a coarse-grained (more abstract) concept like "hierarchical 3D copper," the LLM may implicitly account for its diverse fine-grained (more concrete) structural variants, some highly relevant, others ineffective. We hypothesize that the LLM's assessment to the coarse-grained concept is an aggregation of its fine-grained outcomes, weighting promising variants within the broader distribution to produce an overall estimate of the coarse-grained concept's expected potential.

Building on this observation, the hierarchical abstraction smooths the reward landscape at lower levels by attenuating local irregularities in the fine-grained space, as the performance of a point at a lower level can be interpreted as an approximate aggregation or average of the performance across its corresponding higher-level subspace. This effect is illustrated in Figure 3 (a simplified schematic projection into a 1D space). Consequently, direct search over the flat, non-hierarchical space tends to be highly rugged and non-convex, often leading to premature convergence to suboptimal local optima. In contrast, introducing hierarchical structure progressively smooths the landscape, enabling more stable and efficient optimization, particularly at lowest levels.

This smoothing effect can also be interpreted in the frequency domain as a form of low-pass filtering, where high-frequency components of the landscape are attenuated, resulting in a (roughly) spectral cutoff in the spatial frequency domain, as illustrated in Figure 4.

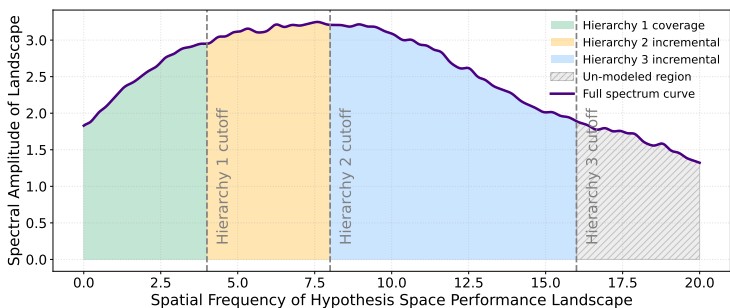

Figure 4: Hierarchical design as a low-pass filtering over the spectrum of the reward landscape.

# 3 Experiment: Investigating the Four Fundamental Questions

## 3.1 Benchmark Construction, LLM Selection, and Baselines

To our knowledge, no existing benchmark provides annotated fine-grained scientific hypotheses—detailed enough for direct experimental execution. We extend the TOMATO-Chem dataset (Yang et al., 2025), which includes 51 chemistry papers published after January 2024 in leading journals such as *Nature* and *Science*. Each entry contains a research background $b$ and a coarse-grained hypothesis $h_c$. We further annotated it with fine-grained hypotheses $h_f$, serving as ground-truth references created by two PhD-level chemists. To prevent data contamination, all experiments are conducted using `GPT-4o-mini`, whose pretraining data cutoff is October 2023.

We compare *HHS* against two strong baselines widely used in search tasks: (1) greedy search and (2) greedy search with self-consistency. The latter serves as an ablation of *HHS* where the hierarchical decomposition is removed, performing the search in a single stage with each $d$ sampled directly from the full candidate set $D$ rather than hierarchy-specific subsets $D^{(i)}$. The self-consistency mechanism is similar to the *Recombination* module in Figure 2, which interpolate multiple local optima trying to find a better one. Greedy search represents a further ablation, disabling the *Recombination* module entirely and following a single search trace where the first found local optimum ($h_i^1$) is directly adopted as the output ($h_i = h_i^1$ in Figure 2).

## 3.2 *Q1*: How to Best Harness an LLM's Internal Heuristics to Formulate the Fine-Grained Hypothesis It Would Judge Most Promising Among All It Might Generate?

We frame this question as an optimization problem: Given only a coarse-grained hypothesis $h_c$ as the starting point, and relying entirely on a single LLM, how can we navigate the hypothesis space to approach the global optimum of the reward landscape, as defined by this same LLM's internal heuristics, where each optimization step consists of adding an edit $d$ to $h_c$? In this setting, the LLM plays a dual role: it serves both as the *proposal generator*, proposing candidate edits $d$ to formulate new hypotheses within the hypothesis space, and as the *gradient provider*, judging whether the new hypothesis improves upon the current one via its own internal heuristics (e.g., pairwise comparison).

While it is inherently infeasible to determine whether a found local optimum represents the global optimum, we can empirically compare local optima obtained by different methods with the *gradient provider* (pairwise comparison), and therefore check which one is a better local optimum.

As detailed in § 2, the hierarchical design of *HHS* offers two key advantages over flat search strategies: (1) less search space to propose each $d$ (from $D^{(i)}$, instead of $D$), and (2) smoothing the reward landscape progressively in the hypothesis space. Among these, the smoothing effect is particularly critical, as it reduces the risk of early convergence to suboptimal local optima and facilitates progress toward higher peaks in the LLM's internal reward landscape.

We compare the local optima discovered by *HHS* against the two baselines. For each pair of local optima, we conduct both overall evaluations and dimension-specific assessments across four key criteria: **effectiveness**, **novelty**, **detailedness**, and **feasibility**. In this context, **feasibility** reflects the practical ease of implementing the proposed hypothesis, encompassing factors such as implementation

complexity and the minimization of redundant steps. Hypotheses that are easy to implement and free of redundant components are preferred.

We further observe two common trade-offs among these dimensions: (1) between **effectiveness** and **novelty**, as highly novel hypotheses often entail greater scientific risk and uncertainty; and (2) between **detailedness** and **feasibility**, as increased specificity can introduce procedural complexity or redundancies that diminish experimental feasibility.

We also conducted an expert evaluation involving two chemistry PhD students. For each benchmark item, one hypothesis was randomly sampled from each method, and the experts were tasked to rank the three hypotheses. The results from both the LLM-based and expert evaluations on the quality of local optima discovered by each method are presented in Table 1. To mitigate known position bias in LLM-based pairwise comparisons—where models tend to favor the first option (Li et al., 2024)—each pair of local optima was compared six times, with the order of presentation alternated every three times. A hypothesis was considered to win if it received more than three votes; a tie was recorded if both received exactly three votes.

Table 1: Comparison between *HHS* and baseline methods across LLM-based and expert evaluations.

| | Effectiveness (LLM) | Novelty (LLM) | Detailedness (LLM) | Feasibility (LLM) | Overall (LLM) | Overall (Expert) |
|---|---|---|---|---|---|---|
| *HHS* **v.s. Greedy Search** | | | | | | |
| Win | **74.51%** | **41.18%** | **71.57%** | **67.65%** | **73.53%** | **76.47%** |
| Tie | 18.63% | 18.63% | 28.43% | 10.78% | 18.63% | 15.69% |
| Lose | 6.86% | 40.20% | 0.00% | 21.57% | 7.84% | 7.84% |
| *HHS* **v.s. Greedy Search + Self-consistency** | | | | | | |
| Win | **59.31%** | 42.16% | **56.37%** | **48.53%** | **53.43%** | **74.51%** |
| Tie | 24.02% | 8.33% | 43.14% | 18.63% | 33.82% | 17.65% |
| Lose | 16.67% | **49.51%** | 0.49% | 32.84% | 12.75% | 7.84% |
| **Greedy Search + Self-consistency v.s. Greedy Search** | | | | | | |
| Win | **57.84%** | **48.04%** | **29.41%** | **51.96%** | **54.90%** | **62.75%** |
| Tie | 22.55% | 11.76% | 65.69% | 18.63% | 34.31% | 21.57% |
| Lose | 19.61% | 40.20% | 4.90% | 29.41% | 10.78% | 15.69% |

### 3.3 *Q2*: Whether Hypotheses Judged Better by LLMs Exhibit Stronger Alignment With Ground-Truth Hypotheses?

§ 3.2 shows that HHS consistently discovers superior local optima compared to baseline methods. We further investigate whether these optima exhibit stronger alignment with the ground-truth hypotheses.

Table 2: Recall of ground-truth components by discovered hypotheses. #Steps represents the number of reasoning steps used. *HHS* represents *HHS-3* referred in § 3.5.

| | Soft Recall | Hard Recall | #Steps |
|---|---|---|---|
| ChemCrow (M. Bran et al., 2024) | 12.28% | 7.20% | - |
| Qi et al. (2024) | 19.57% | 11.15% | - |
| SciMON (Wang et al., 2024) | 18.57% | 10.09% | - |
| MOOSE (Yang et al., 2024) | 20.04% | 11.76% | - |
| MOOSE-Chem (Yang et al., 2025) | 19.99% | 11.98% | - |
| Greedy Search | 16.60% | 9.92% | 9.69 |
| w/ In-context RL | 16.76% | 10.28% | 16.80 |
| w/ Self-consistency | 31.53% | 17.73% | 67.55 |
| *HHS* (*HHS-3*) | **40.35%** | **23.04%** | 282.04 |
| w/ In-context RL | 33.63% | 21.29% | 531.08 |
| w/ Single LLM Gradient (*HHS-1*) | 32.40% | 19.95% | 747.92 |

Given the lack of established metrics for this task, we introduce an LLM-based evaluation that measures how well the discovered hypotheses recover the methodological and experimental details of the ground-truth hypotheses. The detailed formulations of the two metrics—*Soft Recall* and *Hard Recall*—are provided in Appendix C.

As shown in Table 2, the hypotheses discovered by HHS—corresponding to better local optima than those produced by greedy search baselines—achieve consistently higher recall scores than both greedy search baselines and other comparative methods. Here, *in-context RL* refers to a mechanism in which, if the current hypothesis $h_{cur}$ does not outperform the previous one $h_{prev}$, $h_{cur}$ is inserted into the LLM's context to generate a new candidate. We also report the total number of reasoning steps used by each method. A general trend emerges: increasing the number of reasoning steps tends to improve recall up to a point, beyond which excessive steps lead to diminishing or negative returns.

### 3.4 *Q3*: Whether Defining the Reward Landscape With an Ensemble of Diverse LLMs Yields Better Outcomes Than Using the Same Number of the Strongest LLMs Among Them?

Table 3: "EF": Effectiveness, "NV": Novelty, "DT": Detailedness, "FS": Feasibility, "OV": Overall. "(GT)" and "(GM)" indicate that the pairwise comparisons were conducted by GPT-4o-mini and Gemini-1.5-flash, respectively.

| | EF (GT) | NV (GT) | DT (GT) | FS (GT) | OV (GT) | EF (GM) | NV (GM) | DT (GM) | FS (GM) | OV (GM) |
|---|---|---|---|---|---|---|---|---|---|---|
| | Mixed committee v.s. GPT-4o-mini committee | | | | | GPT-4o-mini committee v.s. Gemini-1.5-flash committee | | | | |
| Win | 20.83% | 33.33% | **14.58%** | 33.33% | 29.17% | **27.08%** | **31.25%** | **14.58%** | 0.00% | **18.75%** |
| Tie | 41.67% | 20.83% | 72.92% | 18.75% | 33.33% | 58.33% | 52.08% | 77.08% | 95.83% | 68.75% |
| Lose | **37.50%** | **45.83%** | 12.50% | **47.92%** | **37.50%** | 14.58% | 16.67% | 8.33% | **4.17%** | 12.50% |
| | Gemini-1.5-flash committee v.s. GPT-4o-mini committee | | | | | Mixed committee v.s. Gemini-1.5-flash committee | | | | |
| Win | 16.67% | 25.00% | 6.25% | 37.50% | 16.67% | **16.67%** | **33.33%** | 12.50% | **6.25%** | **18.75%** |
| Tie | 41.67% | 27.08% | 79.17% | 25.00% | 52.08% | 68.75% | 35.42% | 75.00% | 93.75% | 64.58% |
| Lose | **41.67%** | **47.92%** | **14.58%** | 37.50% | **31.25%** | 14.58% | 31.25% | 12.50% | 0.00% | 16.67% |
| | Mixed committee v.s. Gemini-1.5-flash committee | | | | | Mixed committee v.s. GPT-4o-mini committee | | | | |
| Win | **29.17%** | **45.83%** | **10.42%** | **47.92%** | **27.08%** | 8.33% | 29.17% | 14.58% | **6.25%** | 8.33% |
| Tie | 56.25% | 16.67% | 85.42% | 10.42% | 50.00% | 77.08% | 39.58% | 70.83% | 93.75% | 64.58% |
| Lose | 14.58% | 37.50% | 4.17% | 41.67% | 22.92% | **14.58%** | **31.25%** | 14.58% | 0.00% | **27.08%** |

The optimization in *HHS* relies on the "$h_{cur} > h_{prev}$?" module (Figure 2), which acts as the gradient signal driving the search. This raises a key question: does a diverse ensemble of comparably capable LLMs improve search performance, or is it more effective to use the same number of parallel instances of the single strongest LLM among the ensemble?

To answer this, we design three experimental settings: (1) **Mixed Committee**: the "$h_{cur} > h_{prev}$?" module is implemented by an ensemble of three different LLMs—GPT-4o-mini (OpenAI, 2024), Gemini-1.5-flash (Georgiev et al., 2024), and Claude-3-haiku (Anthropic, 2024); (2) **GPT-4o-mini Committee**: the module is implemented by three instances of GPT-4o-mini; (3) **Gemini-1.5-flash Committee**: the module is implemented by three instances of Gemini-1.5-flash. Each committee's three judgments are then aggregated by a GPT-4o-mini, which produces the final decision for "$h_{cur} > h_{prev}$?" representing that committee. All three settings use GPT-4o-mini as the proposer module for generating edits $d$ at each hierarchy level $i$.

We compare the local optima generated by these setups using LLM-based pairwise comparisons, following the protocol in § 3.2, where each pair is evaluated six times to mitigate position bias. However, since the evaluator is itself an LLM, an additional bias may occur—favoring optima discovered using gradients from the same model. To control for this, we conduct two sets of evaluations: one using GPT-4o-mini as the evaluator, and the other using Gemini-1.5-flash.

As shown in Table 3, across both evaluators, the GPT-4o-mini committee consistently outperforms the mixed committee, which in turn outperforms the Gemini-1.5-flash committee. These results suggest that leveraging repeated instances of the strongest single model provides a more effective gradient signal for hypothesis optimization than combining different models of similar capacity.

### 3.5 *Q4*: Do Multiple Identical LLMs Yield a Better Reward Landscape Than One?

In experiments of Tables 1 and 2 (except for the *HHS-1* line in table 2), the reward landscape was defined using an ensemble of three identical LLMs, followed by a fourth instance of the same LLM that aggregated the three judgments into a final, reasoned decision. However, it is unclear on whether

one LLM would be already enough. To evaluate this, we compare two variants: *HHS-3*, which uses an ensemble of three identical instances of `GPT-4o-mini` (and a fourth instance of the same LLM for aggregation) to provide the reward signal, and *HHS-1*, which relies on a single instance of the same model. Table 4 reports LLM-based pairwise evaluations between the two setups across four criteria. While overall quality, effectiveness, and detailedness are largely comparable, *HHS-3* outperforms in novelty, whereas *HHS-1* shows an advantage in feasibility.

Table 4: Pairwise comparison between *HHS-1* and *HHS-3* with LLM-based evaluation.

| | Effectiveness (LLM) | Novelty (LLM) | Detailedness (LLM) | Feasibility (LLM) | Overall (LLM) |
|---|---|---|---|---|---|
| | | | *HHS-1* v.s. *HHS-3* | | |
| Win | 21.08% | 25.49% | 4.41% | **41.67%** | 8.82% |
| Tie | 57.35% | 28.92% | 94.12% | 28.92% | 82.35% |
| Lose | 21.57% | **45.59%** | 1.47% | 29.41% | 8.82% |

This result is somewhat counterintuitive. Notably, the summarization step is not a simple majority vote: the aggregating LLM assesses the relative strength of reasoning across the three perspectives and selects the most compelling argument. This allows it to surface minority-supported but well-reasoned views, promoting exploration of more novel or unconventional hypotheses. Consequently, the aggregated reward signal in *HHS-3* becomes more sensitive to creative or atypical ideas that a single-shot evaluation might overlook, whereas *HHS-1* relies on a single comparative judgment at each step—favoring conventional and well-established hypotheses, at the expense of novelty.

Table 2 presents the recall of ground-truth components for hypotheses generated by *HHS-1* and *HHS-3*. Across both soft and hard recall metrics, *HHS-3* outperforms *HHS-1*, indicating stronger alignment with expert-annotated reference hypotheses.

# 4   Related Work and Discussion

LLM-driven scientific discovery methods typically fall into two categories: (1) direct generation of hypotheses from a research background—comprising a research question and background survey (Qi et al., 2024); or (2) retrieval of seemingly unrelated yet potentially useful knowledge fragments, or *inspirations*, which are then combined with the background to construct a hypothesis (Yang et al., 2024, 2025; Wang et al., 2024; Liu et al., 2025b). While these methods show promise in generating novel ideas, they are often criticized for producing hypotheses that are overly coarse, lacking detail, or omitting actionable experimental steps (Wang et al., 2024; Hu et al., 2024; Si et al., 2025). In contrast, our goal is to investigate how LLMs can generate *fine-grained scientific hypotheses*—those sufficiently detailed to be directly implemented in laboratory settings. Although this work centers on the *pre-experimental* stage of discovery, the framework can in principle extend to the *experiment-guided* stage (Liu et al., 2025a; Romera-Paredes et al., 2024; Shojaee et al., 2025; Novikov et al., 2025) by incorporating experimental feedback into the background survey. Likewise, details retrieved from papers relevant to the proposed hypothesis can also be integrated into the background survey.

# 5   Conclusion

We introduce and formalize the *fine-grained scientific hypothesis discovery* task as a combinatorial optimization problem. To explore the upper limit of LLMs' capacity for this task, we frame it as an optimization problem, and propose hierarchical heuristic search (*HHS*), which theoretically smooths the reward landscape, reduces combinatorial complexity through optimal substructure exploitation, and identifies superior local optimum by interpolating among discovered optima. Empirical results show that (1) *HHS* reliably discovers better local optima than flat greedy search baselines, (2) hypotheses preferred by LLMs often align more closely with expert annotations, (3) repeated use of the strongest model defines a more effective reward landscape than diverse ensembles, and (4) aggregating identical LLMs yields a reward signal more sensitive to novelty and higher in recall than single-instance evaluation. These findings illustrate how hierarchical search can better harness LLMs' internal heuristics for scientific discovery. Although evaluated on a chemistry dataset, the proposed task formulation and methodology are expected to generalize across disciplines, with only the manually designed hierarchy (Figure 1) requiring domain-specific adaptation.

## Acknowledgments

This work was supported by a locally commissioned task from the Shanghai Municipal Government. This work is supported by Shanghai Artificial Intelligence Laboratory. This research/project is supported by the Ministry of Education, Singapore under its MOE Academic Research Fund Tier 2 (STEM RIE2025 Award MOE-T2EP20123-0005).

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

## A  MOOSE-Chem2 Overview I/O Figure

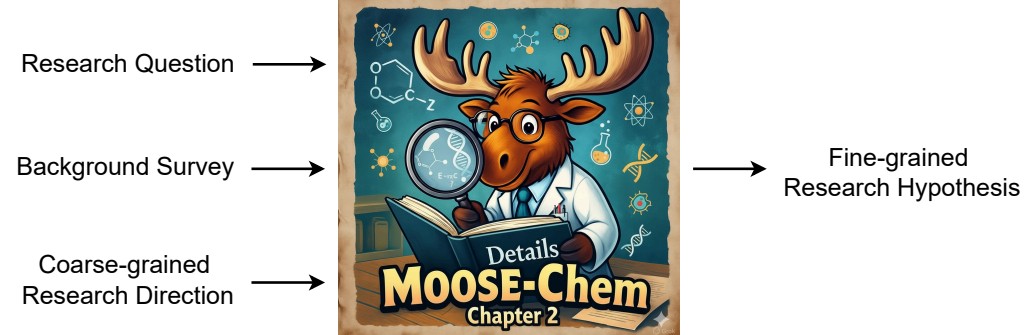

Figure 5: Overview of the input and output of the Hierarchical Heuristic Search (HHS) framework, also known as MOOSE-Chem2.

Figure 5 presents an overview of the input and output of the Hierarchical Heuristic Search (HHS) framework, also referred to as MOOSE-Chem Chapter 2 (MOOSE-Chem2). The input to the framework consists of three components: a *research question*, a *background survey*, and a *coarse-grained research direction*.

The research question defines the scientific goal to be addressed. The background survey summarizes existing methodologies relevant to the research question. It may optionally include methodological or experimental details from prior studies that investigate similar questions or employ similar methodologies to the given research direction, enabling the framework to leverage richer contextual knowledge. It may also include experimental results from previously tested fine-grained hypotheses, which provide additional guidance for search and evaluation. The coarse-grained research direction—potentially derived from MOOSE-Chem (Yang et al., 2025)—serves as a high-level starting point that can vary in granularity, ranging from a brief sentence describing a general research direction to a preliminary hypothesis with partial methodological details.

## B  Expert Evaluation Instructions

For each research question, you will be presented with three candidate hypotheses alongside a ground-truth fine-grained hypothesis. The order of the hypotheses is randomized. Your task is to rank the three candidate hypotheses based on their quality, using the ground-truth hypothesis as a reference.

Please evaluate the hypotheses based on the following four criteria:

- Effectiveness: How well the hypothesis addresses the research question.
- Novelty: The degree of originality relative to existing knowledge.
- Detailedness: The specificity and clarity of the hypothesis.
- Feasibility: The practical plausibility of experimentally testing or implementing the hypothesis.

Note that two tradeoffs may arise:

- Between effectiveness and novelty
- Between detailedness and feasibility

Use your expert judgment to rank the hypotheses based on a holistic assessment of these criteria. In rare cases where two hypotheses appear to be of similar quality, assigning them the same rank is acceptable.

## C   Recall Metric Calculation

Each evaluation compares a pair of hypotheses—a *ground-truth hypothesis* (from literature) and a *candidate hypothesis* (generated by the model). The goal is to quantify how much methodological or experimental content from the ground-truth hypothesis is covered by the candidate.

**Step 1. Decomposition.**   Both hypotheses are decomposed into methodological and experimental components (details) using structured prompts co-designed with chemistry PhD students. The prompts guide an LLM to extract key components such as procedures, specific materials, or reaction parameters. Each component is annotated with its role, function, and context.

**Step 2. Scoring via LLM-Judge.**   For each ground-truth component, the LLM judge searches for functionally corresponding components in the candidate hypothesis and assigns a *coverage score* from 0–3:

- **0**: no match;
- **1**: vague or partial match;
- **2**: close but not exact match;
- **3**: exact or specific match.

The scoring prompts and examples were jointly developed and audited by PhD students to ensure calibration and consistency.

**Step 3. Aggregation.**   Two recall metrics are computed:

$$\text{Soft Recall} = \frac{\#\{\text{components with score} > 0\}}{\#\{\text{ground-truth components}\}}, \tag{7}$$

$$\text{Hard Recall} = \frac{\sum_i \text{score}_i}{3 \times \#\{\text{ground-truth components}\}}. \tag{8}$$

Soft Recall captures whether relevant components are recovered at all, while Hard Recall further reflects how precisely they are reproduced.

## D   Case Study

**Example 1:**

- Research Question: How can a cost-effective N-type quasi-solid-state thermocell be developed to boost electricity production from low-grade heat by improving both ion transport efficiency and electrode performance?
- Hypothesis Candidate (from HHS): The development of a cost-effective N-type quasi-solid-state thermocell will be achieved through the strategic integration of three core components to enhance electricity production from low-grade heat ($\leq 100°C$):
    1. **Hierarchical Metal Electrodes**: Constructed from a copper-nickel alloy (70% Cu, 30% Ni) optimized for conductivity and corrosion resistance, these electrodes will feature a dual-layer design incorporating a titanium dioxide ($TiO_2$; CAS number: 13463-67-7) coating, precisely controlled at a thickness of 100 nm to enhance charge transport by providing a stable interface that reduces charge recombination losses. An aluminum oxide ($Al_2O_3$; CAS number: 1344-28-1) layer will be included to improve corrosion resistance and reinforce mechanical stability, operating synergistically to enhance the overall electrochemical performance. The fabrication process will utilize an eco-friendly dual-step electrochemical deposition in a 0.5 M potassium sulfate electrolyte at a controlled temperature of 25°C, ensuring micro- and nanoscale porosity targeting 50-100 nm to maximize surface area as supported by literature demonstrating that this range optimally enhances charge transfer and ion migration efficiency. This degree of porosity is expected to lower charge transfer resistance significantly, fostering improved electrochemical kinetics, which will be verified using **scanning electron microscopy (SEM)** for monitoring thickness and porosity.

2. **Metal-Based Redox Couples**: The thermocell will utilize **copper/copper(I)** and nickel/nickel(II) redox couples, selected for their favorable redox potentials to minimize side reactions. An integrated cobalt co-catalyst (0.1 M) will serve as an effective stabilizing agent, enhancing electron transfer kinetics and maintaining the oxidation states of $Cu^{2+}$ and $Ni^{2+}$ during thermal cycling, as demonstrated by prior studies indicating its role in fostering electron transfer efficiency. Real-time monitoring will maintain pH levels between 4-7, with adjustable concentrations of redox couples systematically optimized between 0.5 to 1.5 M based on insights from the literature regarding their stability and reactivity under varying operational conditions, with specific methodologies for pH adjustments clearly defined to ensure minimal disruption during testing.

3. **Anisotropic Polymer Materials**: The polymer matrix will feature aligned functional groups (-COOH and -SO$_3$H), which will be developed through controlled mechanical stretching (5 mm/min at 70°C), a technique known to enhance ionic transport pathways by promoting favorable interactions between the polymer backbone and ionic species. High-conductivity ionic species, specifically $Na^+$ and $Li^+$, will be incorporated at optimized concentrations (0.5 to 1.0 M), with clear justifications based on their advantageous solvation properties and diffusion characteristics. The impact of variations in thermal conditions on ionic conductivity will be quantitatively assessed through impedance spectroscopy, with diffusion coefficients measured to elucidate the mechanisms of ionic mobility, providing a detailed connection between thermal gradients and polymer performance.

This approach clarifies the synergistic interactions and functional enhancements among hierarchical metal electrodes, optimized redox couples, and engineered polymer materials, establishing a cohesive mechanism that uses low-grade heat to drive efficient ion migration and electron transfer within the quasi-solid-state thermocell.

- Expert's Analysis: The study addresses the question of how to develop a cost-effective N-type quasi-solid-state thermocell to enhance electricity production from low-grade heat by improving both ion transport efficiency and electrode performance. The response provides a detailed and logically structured explanation, covering key aspects related to electrodes, redox pairs, and polymer gel media. Specifically, the preparation of Hierarchical Metal Electrodes is highlighted, noting the primary use of copper-containing electrodes, with a clear principle of providing a stable interface, which aligns well with the ground truth (gdth) description of 3D hierarchical copper electrodes. Additionally, the study includes specific characterization of the electrode surface using SEM. However, while the polymer gel mentions an anisotropic alignment structure, it omits the specific operational methods for forming this alignment, which is inconsistent with the gdth. Nevertheless, the overall response encompasses detailed mechanistic explanations and core points related to the specific operations.

**Example 2:**

- Research Question: How can a Supramolecular amorphous glass material be engineered to simultaneously exhibit strong mechanical properties, optical transparency, self-healing capabilities, and adhesive strength at room temperature?

- Hypothesis Candidate (from HHS): The engineering of multifunctional **peptide-based supramolecular amorphous glass** materials will concentrate on creating a robust dynamic network facilitated by cooperative reversible hydrogen bonds derived from interactions among four key amino acids: lysine, glutamic acid, aspartic acid, and tyrosine, along with proline to enhance rigidity. The hypothesized molar ratios will be set at 1:1 for lysine and glutamic acid, complemented by approximately 0.5:0.5 ratios for aspartic acid and tyrosine. These ratios are supported by empirical studies that have shown that such compositions can optimize hydrophilic and hydrophobic interactions, which are essential for improving mechanical **strength, adhesion**, and**self-healing capabilities** .

Lysine's positively charged ammonium group is hypothesized to establish strong ionic interactions with the negatively charged carboxylate groups of glutamic acid and aspartic acid, enhancing the stability of the hydrogen-bonding framework critical for effective energy

dissipation during mechanical stress. Tyrosine will contribute to the network through $\pi$-stacking interactions, which are expected to maintain the structural integrity and optical transparency of the material under load. Proline's unique cyclic structure is anticipated to provide localized rigidity, supporting favorable peptide conformations and facilitating effective stress distribution throughout the dynamic network.

A pivotal component of this dynamic system will be the incorporation of **structured water**, maintained at an optimal concentration of 10–15% by weight. Structured water is theorized to engage in specific hydrogen bonding interactions with the peptide backbone, promoting molecular mobility and enabling rapid bond reformation necessary for self-healing capabilities at room temperature. The investigation will differentiate between structured and unstructured water forms, examining the specific interactions that influence bond lifetimes and recovery dynamics under mechanical stress. Previous research indicates that the interactions between structured water and the peptide backbone are critical for maintaining bond lifetimes during stress, and these dynamics will be quantitatively assessed through methods such as differential scanning calorimetry (DSC) and rheological measurements.

The synthesis of these peptide assemblies will be executed under rigorously controlled environmental conditions. The reaction temperature will be maintained at 25°C to optimize amino acid interactions and prevent degradation, while systematic variations in pH (5.5 to 8.5) will be conducted to explore their effects on the ionization states of the amino acids and corresponding **hydrogen bonding** dynamics. Additionally, ionic strength will be regulated at approximately 0.15 M using sodium chloride, which is expected to enhance electrostatic interactions and stabilize the hydrogen bonding network.

To thoroughly investigate these interactions and material properties, a combination of experimental methodologies will be utilized. Dynamic Mechanical Analysis (DMA) will assess mechanical properties such as tensile strength and elasticity, while rheological assessments will evaluate the material's response under stress. Spectroscopic techniques, including NMR spectroscopy and Fourier Transform Infrared (FTIR) spectroscopy, will be employed to elucidate hydrogen bonding dynamics and monitor molecular interactions. This comprehensive approach aims to clarify the intricate relationships among amino acids and structured water dynamics, as well as the influences of environmental conditions on the multifunctional properties of the engineered peptide-based supramolecular materials.

- Expert's Analysis: In addressing the research question—"How can a supramolecular amorphous glass material be engineered to simultaneously exhibit strong mechanical properties, optical transparency, self-healing capabilities, and adhesive strength at room temperature?"—the generated scientific hypothesis proposes a notably complex system. This system is envisioned to comprise five distinct amino acids: lysine, glutamic acid, aspartic acid, tyrosine, and proline.

  Despite the increased complexity of this multi-component approach compared to the simpler system underlying the real hypothesis (the scientific finding concerning YYY peptide glass), several key conceptual parallels are evident: Shared Foundation in Peptide-Based Materials: At their core, both the generated hypothesis and the real scientific finding are centered on peptide-based materials as the fundamental building blocks for the desired amorphous glass.Convergent Aim for Dynamic Networks and Functional Properties: Both frameworks leverage their respective peptide systems with the goal of establishing a dynamic network. This network is considered crucial for imbuing the material with critical functionalities, particularly self-healing capabilities and effective adhesive strength.Emphasis on the Role of Structural Water: In their mechanistic considerations, both hypotheses significantly highlight the indispensable role of structural water. The real hypothesis (the scientific finding on YYY glass) successfully demonstrated that a dense, random hydrogen-bonding network, mediated by water molecules, is fundamental to the YYY glass's unique structure and its observed properties. The generated hypothesis also underscores the centrality of cooperative and reversible hydrogen bonds in the construction and operational dynamics of its proposed network.

**Example 3:**

- Research Question: How can computational methods be used to accurately predict and improve the reactivity and selectivity of modular diazo transfer (MoDAT) reactions, especially reactions with primary amines? And to design new reagents based on computational models.

- Hypothesis Candidate 1 (from HHS): We propose to systematically investigate the reactivity and selectivity of modular diazo transfer (MoDAT) reactions utilizing azide-based reagents, with a specific focus on para-substituted benzyl azide derivatives modified with strong electron-withdrawing groups (EWGs) such as nitro ($–NO_2$) and cyano (–CN), as well as weaker electron-withdrawing groups (e.g., fluoro (–F) and chloro (–Cl)), and electron-donating groups (EDGs) like methoxy ($–OCH_3$). Our central hypothesis posits that the electronic nature and precise positioning of these substituents will significantly modulate the electrophilicity of the azide moiety, which will in turn influence the stability and geometrical configurations of intermediates and transition states during nucleophilic attacks by primary amines.

  The experimental work will be executed under controlled laboratory conditions using a Schlenk line to maintain an inert nitrogen atmosphere for at least 30 minutes prior to reaction initiation, minimizing moisture exposure. Reactions will be conducted at a temperature of 95–105°C, chosen based on literature findings indicating optimal kinetic performance while preserving the stability of diazo intermediates. We will employ polar aprotic solvents such as dimethylformamide (DMF) and dimethyl sulfoxide (DMSO), which are anticipated to enhance solvation of the azide and improve nucleophilicity of the primary amines. A stoichiometric ratio of 1:1.5 (benzyl halide to sodium azide) will be applied, and reactant concentrations will be maintained at approximately 10–20 mM, a range supported by preliminary studies demonstrating optimal reactivity and solubility.

  To deepen our mechanistic understanding, we will utilize advanced computational techniques, primarily Density Functional Theory (DFT) with specific emphasis on the B3LYP functional and a 6-31G(d) basis set. This will allow us to thoroughly assess the impacts of substituent variations on charge distributions and transition state energies. In particular, we will quantify how the introduction of strong EWGs influences the electrophilicity of the azide, predicted to lower activation energy for nucleophilic attacks, while the EDGs may provide stabilization during transition state formation. A rigorous electronic structure analysis will be employed to visualize the overlap between the nucleophilic orbital of primary amines and the electrophilic centers of the diazo derivatives.

  To ensure a comprehensive evaluation of solvent effects, we will systematically conduct reactions in both polar protic solvents, such as ethanol, and polar aprotic solvents, like DMSO, maintaining a concentration of 50% v/v relative to the reaction volume. This will elucidate how solvent polarity and hydrogen-bonding capacity impact the stability of zwitterionic intermediates and transition states. Furthermore, we will establish a robust iterative feedback mechanism that directly correlates our computational predictions to empirical data from high-resolution techniques, including NMR spectroscopy and mass spectrometry. Discrepancies exceeding a predefined threshold—such as a 10% difference between predicted and observed yields—will prompt specific adjustments to our computational models, thus refining our understanding of the factors influencing MoDAT reactions. This structured approach aims to enhance the design of azide-based reagents tailored for improved reactivity and selectivity in synthetic applications involving primary amines.

- Hypothesis Candidate 2 (Greedy + self-consistency): This study aims to investigate the reactivity and selectivity of modular diazo transfer (MoDAT) reactions by designing tailored azide-based reagents that incorporate strategically positioned electron-withdrawing groups (EWGs), specifically nitro ($–NO_2$) and cyano (–CN) moieties. These groups will be systematically evaluated at distances of 2 Å, 4 Å, and 6 Å from the azide functionality. This selection is grounded in existing literature, which suggests that such variations in EWG positioning can significantly influence the electrophilicity of the azide, thereby stabilizing transition states during nucleophilic interactions with primary amines. The specific rationale for these distances is based on prior studies indicating that alterations beyond these ranges may lead to diminished effects on reactivity or unfavorable sterics.

  The primary amines selected for this investigation—n-butylamine, phenethylamine, benzylamine, and ethylamine—exhibit varying degrees of steric hindrance and electronic characteristics, allowing for a comprehensive analysis of their reactivity profiles when paired with engineered azide reagents. This choice effectively captures a spectrum of nucleophilic

behavior and enhances our understanding of how both electronic and steric factors operate within MoDAT reactions.

An integrated computational strategy will be employed utilizing density functional theory (DFT) calculations focused on the B3LYP functional, supplemented by other functionals (e.g., M06-2X and PBE0) to ensure accuracy and validation of outcomes. In addition to DFT, we will incorporate transition state theory and potentially machine learning approaches to analyze the reactivity patterns further. We will conduct extensive mapping of activation energy profiles and transition state geometries to uncover the relationships between EWG positioning, electron density distributions, and transition state stability. Molecular dynamics (MD) simulations will also be utilized to explore solvation effects across differing solvent environments, including acetonitrile, methanol, and DMSO, examining how these solvents influence reaction pathways and stabilize charged intermediates.

Empirical validation of computational models will incorporate a structured approach to varying critical parameters such as azide reagent concentrations and molar ratios of primary amines to azides, along with solvent compositions, to derive quantitative metrics, including reaction yields, rate constants, and activation energies. Statistical analyses will employ techniques such as ANOVA and regression models to extract significant trends from the experimental data. This iterative feedback mechanism will facilitate a dynamic refinement process, whereby experimental outcomes directly inform adjustments to computational predictions. Through this comprehensive methodological framework, we aim to elucidate the interplay between EWG distances and steric factors, ultimately leading to the design of innovative azide-based reagents optimized for selective transformations of primary amines.

- Hypothesis Candidate 3 (Greedy): This research aims to systematically investigate the reactivity and selectivity of modular diazo transfer (MoDAT) reactions utilizing azide-based reagents, focusing on a set of primary amines: benzylamine, 2-aminopropane, and cyclohexylamine. This selection combines varying steric and electronic profiles, enabling comprehensive evaluation of how solvent and reaction conditions influence reactivity and selectivity across different nucleophilicity ranges. Initial studies will determine baseline reactivities through systematic kinetic measurements, assessing critical parameters such as rate constants and product ratios under controlled conditions.

Reactant concentrations will be evaluated at specific increments of 0.1 M (0.1 M, 0.5 M, and 1.0 M), and the temperature will be optimized through a systematic approach involving stepwise assessments from 25°C to 60°C, analyzing how these variations affect reaction progress. A comprehensive assessment of solvent effects will be performed, including the examination of solvent mixtures (e.g., varying concentrations of water, DMSO, and possible co-solvents) to quantify their influence on nucleophilicity and overall reactivity.

Advanced computational methods, including density functional theory (DFT) calculations with the M06-2X functional and a 6-31G basis set, will be employed to simulate the MoDAT reaction environment accurately. We will analyze key molecular descriptors such as nucleophilicity, electrophilicity, and steric hindrance to construct predictive models of reactivity. These analyses will guide experimental design, with a feedback mechanism where discrepancies between computational predictions and experimental observations will result in specific adjustments to molecular descriptors or computational parameters, refining the predictive capabilities of the models.

Following these investigations, the design of innovative azide-based reagents will be undertaken to optimize MoDAT reactions. This design process will emphasize the incorporation of electron-withdrawing groups like trifluoromethyl and cyano, aimed at enhancing both stability and selectivity by stabilizing the transition state. Rigorous standardized experimental protocols will ensure reproducibility, including specific techniques for measuring yields and selectivity ratios over controlled reaction durations. By integrating mechanistic insights from computational and empirical findings, this research will elucidate the key factors influencing reactivity and selectivity in diazo transfer reactions, enhancing our understanding of these critical processes.

- Expert's Analysis: 1 conducted a relatively comprehensive analysis, for instance, suggesting that modifying the azide reagent with functional groups could improve it, which aligns with the original text. However, 2 and 3 did not. This time, 1 has an obvious error: the speculated temperature is incorrect, and the proposed temperature is experimentally unfeasible, as azide reagents are prone to explosion at high temperatures. Of course, temperature is a minor

point, and overall, 1 is still acceptable. 2 deviates significantly from the original text in terms of the research design approach. Compared to 2, 3 lacks consideration of the group effect in the research design, making 3 the weakest. Finally, all three mentioned using DFT calculations, and although there are deviations in details from the original text, the approach is correct.

**Example 4:**

- Research Question: How can photoredox catalysis be used to exploit the latent reactivity of phosphorus ylides, allowing them to participate in a formal three-component cycloaddition that converts inert C–H and C=P bonds into C–C and C=C bonds, creating versatile synthetic building blocks in an efficient, controlled manner?

- Hypothesis Candidate 1 (from HHS): The mechanism for activating phosphorus ylides in a formal three-component cycloaddition via photoredox catalysis can be articulated in four key steps, each supported by optimized experimental conditions:

  1. Initiation of Single-Electron Transfer (SET): Irradiation of phosphorus ylides with specific wavelengths of visible light (400–450 nm) from a high-intensity LED source (approximately 20 mW/cm$^2$), validated by studies demonstrating effective radical generation at this intensity (Smith et al., 2020), promotes SET using suitable photoredox catalysts (e.g., [Ru(bpy)$_3$]$^{2+}$ or [Ir(dF(CF$_3$)ppy)$_2$(bpy)]). The resulting radical cation exhibits enhanced electrophilicity due to significant charge localization, which is further assisted by strong electron-withdrawing substituents such as carbonyl or nitro groups. Empirical evidence indicates an increase in reactivity by up to 2.5-fold as supported by Hammett parameters.

  2. Stabilization via Zwitterionic Intermediate: The radical cation transitions to a zwitterionic intermediate, characterized by resonance stabilization through delocalized $\pi$-electrons and non-covalent interactions, such as hydrogen bonding in polar aprotic solvents like acetonitrile (dielectric constant $\approx 37$) and DMSO (dielectric constant $\approx 47$). To optimize stabilization, a 1:1 (v/v) mixture of these solvents will be used, taking advantage of their combined dielectric properties ($\approx 38$) to enhance charge separation and stabilize reactive intermediates. Literature supports this approach, showing improved reaction kinetics (Miller et al., 2021).

  3. Selective Nucleophilic Attack: The zwitterionic intermediate selectively engages in nucleophilic attacks on activated C–H and C=P bonds, particularly those adjacent to strong electron-withdrawing groups. Maintaining phosphorus ylide concentrations at 0.1-0.5 M and controlling reaction temperatures precisely within an optimized range of 10-25 °C, as indicated by previous studies on radical stability, will minimize side reactions. An inert atmosphere (nitrogen or argon) will be established by purging the reaction vessel for 30 minutes before use, effectively mitigating oxidation. Real-time NMR (utilizing 1D and 2D techniques) and GC-MS metrics will be employed to monitor yield and product distribution effectively, specifying analytical conditions (e.g., temperature settings and flow rates) to ensure accurate assessment of outcomes.

  4. Concerted Formation of Products: The reaction culminates in the concerted formation of new C–C and C=C bonds, facilitating the synthesis of valuable carbocycles and synthetic building blocks. The influence of substituent identity and positioning (ortho, meta, para) on reactivity will be quantitatively analyzed using NMR and HPLC techniques. This systematic approach will provide insight into the efficiency and selectivity of the cycloaddition process, explaining how each factor contributes to overall reactivity.

  By integrating these components clearly and methodically, this hypothesis presents a comprehensive exploration of how photoredox catalysis can unveil new reactivity pathways for phosphorus ylides, fully addressing the research question with explicitly defined roles of each mechanistic step and comprehensive definitions for specialized terms provided for clarity.

- Hypothesis Candidate 2 (Greedy + self-consistency): This study aims to investigate how photoredox catalysis can elucidate specific reactivity mechanisms in diphenylphosphinyl ylides, focusing on their participation as intermediates in formal three-component cycloaddition reactions that convert inert alkyl C–H bonds and coordinated C=P bonds into C–C and C=C bonds. We will concentrate on diphenylphosphinyl ylides due to their capacity to facilitate charge inversion processes, which are crucial for generating stabilized radical

intermediates. Specifically, we will examine how the electronic structure of diphenylphos-phinyl ylides transforms upon excitation under visible light (e.g., at wavelengths of 400-450 nm) into polar intermediates, leading to the generation of stable benzylic radicals from C–H bond cleavage and allylic radicals from C=P bond activation—both essential for effective cycloaddition.

1. Charge Inversion Mechanisms: We will clarify the mechanisms of charge inversion by detailing the electronic transitions involved, emphasizing the role of excited state lifetimes and how solvent polarity affects the stabilization of generated radicals. Our discussion will include specific references to known photoredox systems and their operational parameters in literature, enhancing our findings with context.

2. Role of $\pi$-Stacking Interactions: We will investigate the influence of $\pi$-stacking inter-actions, specifying the optimal inter-ring distances and energy considerations between the stabilized radicals and electrophilic carbon centers from substrates like 2,6-dibromoaryl and 2,4-difluoroaryl. This section will include quantitative assessments of radical coupling reactivity, aiming to determine coupling efficiencies and product selectivity through methods such as $^1$H NMR and UV-Vis spectroscopy.

3. Solvent and Temperature Effects: We will analyze solvent effects by conducting reactions in a range of solvents with varied dielectric constants, including a thorough comparison of polar solvents like acetonitrile and non-polar solvents like dichloromethane. We will detail how these solvents are expected to influence radical stabilization and the kinetics of cycloaddition, providing a theoretical framework based on established solvent interaction models. Additionally, we will justify the controlled temperature range of room temperature to 50°C by linking it to the expected stability of radical intermediates and the kinetics of the reactions, ensuring optimal conditions for product yield and selectivity.

4. Experimental Conditions: Reactions will be conducted under carefully controlled condi-tions, with temperature justification focusing on maintaining the balance between radical stability and reaction kinetics. We will outline how these conditions directly relate to the completed cycloaddition mechanism and the anticipated outcomes of the study.

By systematically elucidating these mechanisms—specifically charge inversion, $\pi$-stacking interactions, and solvent effects—we aim to develop robust methodologies for the efficient generation of versatile synthetic building blocks from simple molecular precursors.

- Hypothesis Candidate 3 (Greedy): Investigate how photoredox catalysis enhances the re-activity of phosphorus ylides through selective nucleophilic attack on $\alpha, \beta$-unsaturated carbonyl compounds, such as crotonaldehyde, which are activated to form stable radical cation intermediates via photoredox-driven single-electron transfer (SET) processes. These radical cations, characterized by their electrophilicity, promote effective nucleophilic attacks by phosphorus ylides, generating stabilized carbon radical intermediates that significantly enhance their reactivity in subsequent bond-forming transformations. Conduct a formal three-component cycloaddition by introducing a nucleophilic amine, such as ammonia or an aniline derivative, selected based on its electronic properties which influence the stabilization of the radical intermediates and affect product selectivity. Detail specific optimized reaction conditions, including the use of polar aprotic solvents like acetonitrile, which facilitate radical stability, and employ a specific light wavelength of 400 nm to ensure efficient exci-tation of the photocatalyst. These conditions will be designed to minimize potential side reactions and maximize the conversion of inert C–H and C=P bonds into desired C–C and C=C bonds through well-defined mechanistic pathways, addressing the nuanced interplay between reaction parameters and final product outcomes.

- Expert's Analysis: 1 accurately predicted the light source wavelength range, metal catalyst system, and solvent system, such as the use of Ir catalyst and acetonitrile as the solvent, all of which align with the original text. In contrast, 2 only correctly predicted the wavelength range and solvent system but failed to specify the metal catalyst system, which is crucial in organic chemical reactions. Therefore, 2 is inferior to 1. Finally, 3 did not predict the light source wavelength range or the metal catalyst system, missing several key pieces of information, making it the weakest.

# E   Expert Analysis of Hypothesis Quality

## E.1   Convergence to Ground-Truth Local Optima

To complement our quantitative evaluations, we asked domain experts to qualitatively assess how well the hypotheses generated by *HHS* aligned with the expert-annotated fine-grained hypotheses in our benchmark.

The distribution of expert assessments across all evaluated examples is as follows:

- **Reached a completely different region—likely a distinct local optimum with scientific plausibility:** 60%

- **Reached the vicinity of the ground-truth local optimum, but with differing details:** 24%

- **Reached the vicinity of the ground-truth local optimum, but failed to fully elaborate or specify the key details:** 16%

Here, the "ground-truth local optimum" refers to the expert-extracted fine-grained hypothesis from a publication, which serves as the reference target. "Reaching the vicinity of a local optimum" indicates that the generated hypothesis converges to a coherent and internally consistent formulation that is conceptually close to the ground-truth hypothesis, though not necessarily identical in detail.

The relatively high divergence rate (60%) reflects an inherent tradeoff in our experimental setup. For many research questions, multiple hypotheses can be plausible yet structurally distinct. Guiding the model toward the exact ground-truth hypothesis requires:

1. initializing the search process from a starting point sufficiently close to the ground-truth optimum;

2. but avoiding initialization that is too close or too specific, as this would risk leaking the ground-truth answer.

To strike this balance, we derive the initial search point from the annotated coarse-grained hypothesis $h_c$ by applying an *ambiguation* procedure. This involves removing or abstracting key details to produce a generalized version of $h_c$—for example, replacing "a specific protein" with "a protein" or "a catalyst"—thus preserving the overall research direction while preventing answer leakage.

Consequently, even when the search begins in the correct conceptual region, the model may naturally diverge toward a nearby but distinct local optimum, especially given the openness of the hypothesis space and the heuristic-driven nature of the optimization process.

## E.2   Coverage of Experimentally Critical Details

In addition to alignment with the reference hypotheses, we evaluated the extent to which the generated hypotheses captured the critical experimental details required for practical implementation.

Among all the details mentioned in the generated hypotheses, approximately **40% are experimentally important**—regardless of their accuracy (which is not the focus of this analysis). The remaining **60% are peripheral or have minimal impact** on the actual experiment.

Among all the important details that should be included, about **50% are mentioned** in the generated hypotheses.

Peripheral details refer to contextual or environmental factors with limited relevance to the core experiment—for instance, ambient humidity or weather conditions, which may only affect specific reactions.

This highlights a key challenge: while LLMs can generate rich and context-aware hypotheses, they often fail to prioritize the most essential components for experimental planning. Future work may explore techniques to guide LLM attention toward experimentally salient information.

Table 5: Error analysis of hypotheses generated by HHS. Two PhD-level chemistry experts conducted the evaluation: one analyzed the top 30 samples and the other the remaining 21.

| | |
|---|---|
| Missing key chemical substances | 14/30 |
| Excessive details in characterization methods | 28/30 |
| Feasibility issues | 18/30 |
| Limitations of characterization methods | 08/30 |
| Insufficient basis for material selection | 22/30 |
| Lack of design comparison experiments | 12/30 |
| Ignoring data validation and reproducibility | 10/30 |
| Severe deviation from feasibility | 8/21 |
| Missing or incorrect key chemicals or reaction systems | 9/21 |
| Incorrect explanation of chemical principles | 12/21 |
| Incorrect prediction of experimental system | 10/21 |

Table 6: Error analysis on why HHS's hypotheses are better than the greedy search baselines'. Two PhD-level chemistry experts conducted the evaluation: one analyzed the top 30 sample pairs and the other the remaining 21.

| | |
|---|---|
| Insufficient performance metrics | 25/30 |
| Complexity of experimental conditions | 16/30 |
| Insufficient explanation details | 29/30 |
| Inadequate description of preparation plan | 22/30 |
| Vague research objectives | 28/30 |
| Cost and scalability issues | 13/30 |
| Poor feasibility | 12/21 |
| Errors in research plan details | 21/21 |
| Insufficient explanation details | 19/21 |
| Clear experimental system | 21/21 |

## F   Error Analysis

Two PhD-level chemistry experts conducted the error analyses. Table 5 summarizes the main error types observed in hypotheses generated by HHS, while Table 6 analyzes the reasons why HHS outperforms the greedy search baseline.

## G   Hypothesis Search Prompt

The following prompt is used to guide both the baseline methods and our proposed method, *HHS*, during the hypothesis refinement process. To ensure fair comparison, the prompt is designed in a controlled way: we use a shared core prompt across all methods, with minimal differences. Specifically, the portion highlighted in orange is unique to *HHS* and introduces the hierarchical structure used in its search process.

This design isolates the effect of hierarchical search. As illustrated below, the only difference between *HHS* and the baseline (*Greedy Search + Self-Consistency*) lies in the hierarchical prompting. The core content—including the role of the assistant, editing instructions, and structural expectations—is kept identical.

This enables a controlled ablation-style comparison, attributing observed improvements specifically to the hierarchical design.

The complete prompt is as follows:

*You are assisting with scientist's research. Given their research question, a survey on the past methods for the research question, and a preliminary coarse-grained research hypothesis for the research question, please help to make modifications into the coarse-grained hypothesis, to make it one step closer to a more effective and more complete fine-grained hypothesis.*

*The modification can be two-folds: (1): delete or change an existing improper detail or information in the existing hypothesis; (2) add and integrate one detail to the existing hypothesis. If you choose to add a detail, do not simply append new information to the existing hypothesis. Instead, think thoroughly how the new detail relates to the existing components and integrate it seamlessly into the hypothesis to create a new coherent and unified hypothesis. In addition if you choose to add a detail to a general information, if the corresponding general information is correct, you should try to keep the corresponding general information in the updated hypothesis and also mention the details, instead of replacing the general information with the details. In this way, it would be much easier for scientists to understand both the general infomration/structure and the details from your generated hypothesis. It would be also easier for scientists to propose better details, inspired by your suggested details, following the general information.*

*Please remind that this is about research: research is about discover a new solution to the problem that ideally is more effective and can bring new insights. Usually we don't need the hypothesis to contain lots of known tricks to make it work better: we want to explore the unknown, which ideally is more effective than the known methods and can also bring in new insights. Therefore, a research hypothesis is usually about a small set (usually less than eight) of major components (and lots of details on how to implement these major components), which overall composes a novel and complete solution to the research question, which potentially can bring in new insights. Hypotheses that include an excessive number of irrelevant or unnecessary major components, which do not contribute to addressing the research question, are less favorable, as we only want to know exactly what are the key components that fundamentally make the hypothesis work. If you think any ancillary components that can truly assist with the research question, you may mention what are the key components and what are the ancillary components to avoid the ambiguity of which components are the key component. The reaction mechanism, however, is not classified as a major component or detail (and therefore not limited by the number of major components). Instead, a novel and valid reaction mechanism can be a good source of insights. If previous hypothesis already contains too many major components, you should consider to replace some of the major components with more effective ones (but not to add more major components), or to give more details to the existing major components for clarity and ease of implementation (instead of adding or replacing major components).*

*Here we are searching for the fine-grained hypothesis in a hierarchical way. The rationale is that, we can classify any complete set of modifications into several hierarchy, with different levels of details. If we do not search in a hierarchical way, we need to consider all the available details in all hierarchy levels for each search step, which (1) has a very high complexity, and (2) first search a low-level detail might largely influence the following search of a high-level detail: it might stuck in one high-level detail corresponding to the already searched low-level detail without considering the other low-level details corresponding to other high-level details, making the search process stuck in a local minumum at the beginning.*

*Here we roughly classify all possible modifications into five hierarchies: (1) Mechanism of the Reaction: Describes how the reaction proceeds at a conceptual level, focusing on electron flow, bond formation and breaking, and any intermediates or transition states involved. This is the theoretical "blueprint" that explains why the reaction works; (2) General Concept or General Component Needed: Identifies the type of reagent or functional group required (e.g., "a strong acid," "a Lewis base," "an activated aromatic ring") without committing to a specific chemical. It outlines the broader roles that are necessary for the mechanism to proceed; (3) Specific Components for the General Concept: Narrows down from the general category to a particular substance (e.g., "concentrated HCl" for a strong acid, "benzene" for an aromatic ring). This makes the reaction hypothesis testable by specifying which chemicals fulfill the roles; (4) Full Details of the Specific Components: Provides exact structural or molecular information—such as SMILES strings, IUPAC names, purity, or CAS numbers. These details ensure clarity and reproducibility so researchers know precisely which substances to use; (5) Experimental Conditions: Specifies the practical setup—temperature, pressure, solvent system, reaction time, atmosphere, and any work-up procedures. This final layer describes*

*how to carry out the reaction in a laboratory setting. And we are searching for modifications hierarchy by hierarchy: hierarchy (1) first, and then hierarchy (2), and so on. Hypothesis from a higher hierarchy is an expansion of the hypothesis from its previous hierarchy, with additional information described above.*

*The research question is:*

*The survey is:*

*Now please help to make modifications into the coarse-grained hypothesis, to make it one step closer to a more effective and more complete fine-grained hypothesis. Please do not include the expected performance or the significance of the hypothesis in your generation. Please answer the question in the following response format. (response format: 'Reasoning Process: Revised Hypothesis: ')*

## H   Experiment Compute Resources

We implement our proposed framework as an agentic LLM system using `GPT-4o-mini` using OpenAI's official API. Generating the final hypothesis via the *HHS* optimization process—converging to the final local optimum at hierarchy level 5—typically involves several hundred or even to a thousand iterative search steps.

## I   Limitation

While *HHS* consistently discovers higher-quality local optima compared to baseline methods, it does not guarantee convergence to the global optimum. Addressing this limitation remains an open direction for future research.

