# OpenReview forum: "MOOSE-Chem2: Exploring LLM Limits in Fine-Grained Scientific Hypothesis Discovery via Hierarchical Search"
_NeurIPS.cc/2025/Conference — NeurIPS 2025 poster_

### Official Review · Reviewer_SXMM · 2025-06-17

**Clarity:** 2
**Significance:** 2
**Originality:** 2
**Rating:** 3
**Confidence:** 5

**Summary:**

This study reports a method for scientific hypothesis generation that refines hypotheses in stages, from abstract to concrete, thereby enabling the generation of hypotheses that go beyond coarse-grained ones to include methodological and experimental details. The authors empirically validated that this hypothesis generation approach can produce better hypotheses and more closely align with ground-truth hypotheses. Furthermore, they found that the iterative application of a powerful LLM yields better results than using an ensemble of diverse LLMs.

**Questions:**

Am I correct in understanding that the core contribution of this paper lies in the design of prompts for the LLM and the method of combining its inputs and outputs? Furthermore, is the notion of “combinatorial search” intended as a conceptual framing rather than a literal explicit implementation of search space and search algorithm but rather leveraging prompt-based control of the LLM?

**Ethical Concerns:**

["NO or VERY MINOR ethics concerns only"]

**Final Justification:**

Considering the newly added comparisons with particularly strong baselines, I would like to raise my score. That said, I believe there is still room for improvement in the technical novelty and contributions presented in this paper, and therefore I will maintain my overall negative stance toward acceptance.

**Limitations:**

The experiments could be made better by incorporating stronger benchmarks, more diverse datasets, and a larger number of annotators, and more.

**Quality:**

2

**Strengths And Weaknesses:**

The use of large language models (LLMs) for hypothesis generation in scientific research represents a highly significant and timely challenge. While substantial breakthroughs are expected in the near future, there remain numerous unresolved issues at present. As the authors of the paper under discussion rightly point out, one such issue is that naively prompting LLMs to generate hypotheses often results in overly abstract ideas. Bridging the gap between such abstract hypotheses and their concrete implementation remains an open research problem. The authors’ attempt to address this critical issue of “hypothesis concretization” is a commendable aspect of the work.

Throughout the paper, the authors strive to enhance the reader’s understanding by associating key concepts with mathematical formulations and by employing various analogies. Their approach of interpreting LLM-based generation as a process akin to search and building their methodology around this analogy is both novel and intellectually stimulating.

In terms of evaluation, the authors go beyond relying solely on automated LLM assessment by incorporating human evaluations, which strengthens the credibility of their findings. Moreover, their focus on evaluating hypotheses in terms of their “detailedness” is an insightful and original perspective that adds value to the study. Finally, the proposed method consistently outperforms baseline approaches, demonstrating its practical effectiveness and further suggesting the potential significance of this contribution.

While this is an excellent piece of work that contributes meaningfully to the field of AI-driven hypothesis generation, I believe that further improvements are necessary for it to meet the high standards required for acceptance at NeurIPS. Below, I outline the key reasons for this assessment.

If my understanding is correct, what this paper essentially proposes is the following:
	1.	Iteratively generate hypotheses while evaluating their quality using LLMs as judges,
	2.	Repeat this process multiple times and treat the combined outputs as the hypothesis at a given level of abstraction,
	3.	Use the high-level hypotheses obtained so far to move on to generate more concrete hypotheses (i.e., return to step 1).

To my knowledge, iteratively improving LLM outputs, generating a final output from multiple candidates, and progressively refining outputs into more concrete forms are not particularly novel ideas in and of themselves. In that sense, I felt that the method proposed in this study may not be sufficiently innovative or unique to be considered a significant advancement. The annotation of fine-grained hypotheses added to the dataset is indeed a contribution, but since it is based on an existing dataset, it is not an entirely new dataset.

The authors do not propose a new training method or architecture for LLMs but rather use existing models via APIs. Therefore, one would generally expect a carefully designed method for combining these models or crafting prompts; however, I found these aspects to be somewhat lacking. For these reasons, I believe there is still room for improvement in terms of originality.

This paper reports that the proposed method consistently outperforms baseline approaches, which suggests that the direction of the method could be practically valuable. However, the baselines used for comparison are very basic—essentially just self-consistency. Numerous approaches to hypothesis generation and prompting LLMs have been proposed previously, and it is reasonable to assume that many outperform simple self-consistency. Therefore, it was difficult to assess how well this method compares to more sophisticated alternatives.

Of course, the comparisons with the chosen baselines are appropriate for validating the hypothesis, but to more convincingly demonstrate the utility and value of this approach to the AI community, I believe it would be beneficial to include comparisons with stronger baselines.

Regarding the experimental design, it is important to evaluate alignment with ground truth, but in this case, such evaluation was conducted using LLMs as judges. It was unclear what specific instructions were given to the LLM judges or how effective they were—for example, how well their evaluations aligned with human judgments.

Human evaluations were conducted for the baseline comparisons, which I find desirable. However, the results only reported whether the proposed method outperformed existing methods or not. It was not clear how desirable the method’s properties were on their own, independent of its relative performance. While itemized evaluations are provided, only the overall human evaluation scores were reported. As a result, we cannot tell how each specific evaluation criterion was judged by human researchers. Therefore, I was not fully convinced of the overall persuasiveness of the experimental claims.

Finally, while the use of metaphors and concept diagrams throughout the paper made the authors’ ideas more understandable, I felt that these descriptions were somewhat excessive in light of the need for more detailed explanations of the methodology, results, and additional analyses. For example, while the authors carefully explain that the process can be interpreted as combinatorial search, the actual proposal is just a particular way of prompting and combining outputs from an LLM—not a method involving any explicit combinatorial search technique. If that’s the case, I believe the paper would benefit more from devoting space to explaining the methodological details more concretely, as this would better convey the specifics of the research to readers.

---

> ### Author Rebuttal · Authors · 2025-07-31
>
> We are grateful for the detailed comments. Here, we present our responses corresponding to the questions you've asked.
>
> ---
>
> **Q1**: what this paper essentially proposes, and what is the core contribution of this paper.
>
> **A1**: The core contribution of this paper is the formalization of the novel fine-grained hypothesis discovery task as a combinatorial optimization problem, along with the development of Hierarchical Heuristic Search (HHS) to optimize this task effectively for scientific discovery. To our knowledge, no prior LLM-based scientific discovery work has framed the task through an optimization lens. Furthermore, HHS adapts coarse-to-fine search methods to the challenging, language-based open-ended tasks like hypothesis generation, an approach that was previously unclear.
>
> Specifically, we propose a novel framing of the task that enables the use of LLMs to provide the gradient for optimization. To enhance optimization, we introduce a hierarchical design that smooths the reward landscape, facilitating convergence to better local optima. Additionally, we adopt evolutionary search to interpolate among multiple local optima, further refining the optimization process.
>
>
>
> **Q2**: The authors do not propose a new training method or architecture for LLMs but rather use existing models via APIs.
>
> **A2**: The proposed method is essentially a test-time scaling approach, where computation is used as a trade-off for improved performance. This represents a cutting-edge research direction. For example, OpenAI’s Dan Roberts discussed in his talk “9 Years to AGI? OpenAI’s Dan Roberts Reasons About Emulating Einstein” that an ideal scientific discovery system might require 8 years of inference-time computation to propose a high-quality hypothesis, emulating Einstein's process in developing the theory of relativity.
> The key challenge is designing a scaling method that effectively leverages large amounts of compute (e.g., inference-time or API usage) to improve performance. While we currently have access to significant compute resources, the question remains how to best utilize them for generating high-quality research hypotheses. Our method makes a meaningful step in this direction.
>
>
> **Q3**: Although the proposed method consistently outperforms baseline approaches, the baselines used for comparison are very basic—essentially just self-consistency. Better to compare with previous proposed methods.
>
> **A3**: Thank you for the suggestion to compare with additional baselines. While previous works have addressed hypothesis generation, they do not specifically focus on fine-grained hypothesis discovery. We initially chose not to include them as baselines because they would not offer comparable performance in terms of providing detailed hypotheses.
> We have since added experiments with five strong baselines for comparison, with the newly added baselines highlighted in bold.
>
>
> |                            | Soft Recall | Hard Recall |
> |----------------------------|-------------|-------------|
> | **ChemCrow[1]**               |      12.28% |       7.20% |
> | **Reinforcement Learning** |      16.76% |      10.28% |
> | **[3]**                     |      19.57% |      11.15% |
> | **MOOSE[2]**                  |      20.04% |      11.76% |
> | **MOOSE-Chem[4]**             |      19.99% |      11.98% |
> | Greedy                     |      16.60% |       9.92% |
> | Greedy + Self-Consistency                         |      31.53% |      17.73% |
> | HHS                         |      40.35% |      23.04% |
>
> [1] ChemCrow: Augmenting large-language models with chemistry tools, Nature 2023
>
> [2] Large Language Models for Automated Open-domain Scientific Hypotheses Discovery, ACL 2024
>
> [3] Large language models are zero shot hypothesis proposers, COLM 2024
>
> [4] MOOSE-Chem: Large Language Models for Rediscovering Unseen Chemistry Scientific Hypotheses, ICLR 2025
>
>
> **Q4**: Regarding the experimental design, it is important to evaluate alignment with ground truth, but in this case, such evaluation was conducted using LLMs as judges. It was unclear what specific instructions were given to the LLM judges or how effective they were—for example, how well their evaluations aligned with human judgments.
>
> **A4**: Thank you for your comment. Due to word limitations, we are unable to provide the full prompts here, but they are open-sourced and can be found in the evaluation_instruction_prompts() function in ./Method/utils.py.
>
> We worked closely with chemistry experts during the development of the evaluation framework to ensure its quality. A significant portion of our research was dedicated to developing a robust evaluation process.
>
> Additionally, we performed an analysis to assess the consistency between LLM judges and human expert evaluations. Specifically, 100 hypotheses were sampled and evaluated by two human experts (each reviewing 50). The experts were satisfied with 97 out of 100 evaluations.
>
>
>
> **Q5**: Human evaluation results are only reported whether the proposed method outperformed existing methods or not. It was not clear how desirable the method’s properties were on their own, independent of its relative performance.
>
> **A5**: Thank you for the suggestion. Table 2 in the paper presents the recall for each method, which reflects absolute performance rather than relative performance.
>
> Additionally, we conducted further error analyses performed by human experts to gain a deeper understanding of the experimental results using very strict criteria to understand the absolute performance of HHS.
>
>
> **Analysis**:
> Error analysis on the proposed method, HHS. Two chemistry PhD candidates conducted it, when one analyzed 30 hypotheses in terms of 30 research questions, the other analyzed another 21 hypotheses in terms of another 21 research questions. The results are as below:
>
> | HHS Error Analysis                                     | Count |
> |--------------------------------------------------------|-------|
> | Missing key chemical substances                        | 14/30 |
> | Excessive details in characterization methods          | 28/30 |
> | Feasibility issues                                     | 18/30 |
> | Limitations of characterization methods                | 08/30 |
> | Insufficient basis for material selection              | 22/30 |
> | Lack of design comparison experiments                  | 12/30 |
> | Ignoring data validation and reproducibility           | 10/30 |
> | Severe deviation from feasibility                      |  8/21 |
> | Missing or incorrect key chemicals or reaction systems |  9/21 |
> | Incorrect explanation of chemical principles           | 12/21 |
> | Incorrect prediction of experimental system            | 10/21 |
>
>
> **Q6**: While itemized evaluations are provided, only the overall human evaluation scores were reported. As a result, we cannot tell how each specific evaluation criterion was judged by human researchers.
>
> **A6**: Thank you for the suggestion. We have conducted an additional analysis by human researchers for each specific evaluation criterion. The results are provided below:
>
> |                                                       | Effectiveness | Novelty | Detailedness | Feasibility |
> |-------------------------------------------------------|---------------|---------|--------------|-------------|
> |      HHS v.s. Greedy   Search  + Self-consitency      |               |         |              |             |
> | win                                                   | 52.38%        | 47.62%  | 52.38%       | 52.38%      |
> | Tie                                                   | 42.86%        | 42.86%  | 47.62%       | 42.86%      |
> | Lose                                                  | 4.76%         | 9.52%   | 0.00%        | 4.76%       |
> |                HHS v.s. Greedy   Search               |               |         |              |             |
> | win                                                   | 61.90%        | 61.90%  | 66.67%       | 61.90%      |
> | Tie                                                   | 38.10%        | 38.10%  | 33.33%       | 38.10%      |
> | Lose                                                  | 0.00%         | 0.00%   | 0.00%        | 0.00%       |
> | Greedy Search +   Self-consitency  v.s. Greedy Search |               |         |              |             |
> | win                                                   | 47.62%        | 42.86%  | 47.62%       | 47.62%      |
> | Tie                                                   | 42.86%        | 42.86%  | 42.86%       | 42.86%      |
> | Lose                                                  | 9.52%         | 14.29%  | 9.52%        | 9.52%       |
>
>
> **Q7**: While the authors carefully explain that the process can be interpreted as combinatorial search, the actual proposal is just a particular way of prompting and combining outputs from an LLM—not a method involving any explicit combinatorial search technique.
>
> **A7**: In line 122 we analyze that one of the challenges for the fine-grained hypothesis discovery task is that the full search space is unknown. We deal with this challenge by adopting heuristic search in each hierarchy instead of precise search over the full space, which can avoid the need to know the full search space. We use LLMs for this heuristic search, which is prompting looks from the outside.
>
> Out of the classic combinatorial search method, MCTS and genetic algorithms require a clearly defined search space, making them not suitable for this task. Reinforcement learning, however, does not require a defined search space, and we have implemented it as an additional baseline shown in the Table of A3.
>
> ----
>
> *We greatly appreciate your comprehensive feedback. We hope that our responses have satisfactorily addressed all your queries. Should you have further questions or suggestions for enhancing our manuscript, we warmly welcome your input.*

---

> > ### Comment · Reviewer_SXMM · 2025-08-04
> > **Response to authors**
> >
> > Q1
> > If I understand correctly, it seems that the method is not actually differentiating the loss landscape to compute gradients, but rather using the LLM to generate reward-like signals that help guide improvement. In that sense, I would view this as more of an LLM-based evaluation mechanism, rather than the LLM providing gradients per se.
> >
> > I don’t dispute the idea of framing the task as a combinatorial optimization problem, and that may indeed be novel. However, it’s not entirely clear to me how this framing has substantively contributed to devising new method in a useful way.
> >
> > ⸻
> >
> > Q2
> > There is no issue with using APIs. My point was rather that, given the method doesn’t introduce a new model architecture, training procedure, or dataset, the design choices relying on the API are expected to offer a high degree of novelty or utility. I’m not fully convinced that the method reaches that level yet.
> >
> > ⸻
> >
> > Q3
> > I do think the strong performance against baselines like ChemCrow and MOOSE is an important and promising result. Thank you for providing the additional comparative analysis.
> >
> > ⸻
> >
> > Q4
> > Thank you very much for the clarification on the prompts and for the additional explanation regarding the evaluation. Since the evaluation process is such a critical component of this work, I would strongly recommend including more details in the paper itself. Doing so could make the work even more compelling to readers.
> >
> > ⸻
> >
> > Q5
> > Apologies—my earlier comment may have been inaccurate. The table you’ve kindly shown here looks different from Table 2 in the paper. Could you clarify where this version appears?
> >
> > ⸻
> >
> > Q6
> > This is excellent—thank you. Including these more detailed human evaluations definitely makes the work more persuasive and well-rounded.
> >
> > ⸻
> >
> > Q7
> > Thank you. I think the inclusion of reinforcement learning as an additional baseline—as it does not require an explicitly defined search space—helps to illustrate the usefulness of your approach more clearly. Adding such comparisons definitely strengthens the case for your method.

---

> ### Author Response · Authors · 2025-08-04
>
> Thank you so much for the reply.
>
> **Q1**: If I understand correctly, it seems that the method is not actually differentiating the loss landscape to compute gradients, but rather using the LLM to generate reward-like signals that help guide improvement. It’s not entirely clear to me how this framing has substantively contributed to devising new methods in a useful way.
>
> **Response**:
> Thank you for this important question. There may be a misunderstanding about our approach that we'd like to clarify.
>
> Our method does navigate a reward landscape relying on "gradients" for optimization, but within a hypothesis space rather than a continuous parameter space. Specifically:
>
> **Conceptual Framework:** We conceptualize a hypothesis space where each point along the input dimensions (the x-axis, potentially multidimensional) represents a candidate hypothesis, and each point is assigned a reward value (on the y-axis) by the LLM based on its internal heuristics. This defines a reward landscape over the hypothesis space, with the highest peak corresponding to the hypothesis the LLM internally judges as most promising (as described in line 51~55).
>
> **Gradient Computation:** At each step, we incorporate an addition/deletion/replacement of detail to update the hypothesis in the previous step (Equation 6). The pairwise comparison between the previous hypothesis and the modified version provides the optimization direction—essentially "computing" the "gradient" in hypothesis space. This pairwise evaluation serves both as assessment and gradient estimation.
>
> **Methodological Contributions:** This framing enables us to:
>
> 1. Explain why our hierarchical method would lower the complexity (Sections 2.3, 2.6).
> 2. Explain why we choose to adopt heuristic search to further lower the complexity to make the task practical.
> 3. Identify that the essential task is to incorporate a set of details to the initial search point (a very coarse-grained research direction). We frame this incorporation sequentially (Equation 6 in the paper), with each step incorporating one or a few additions/modifications.
>
> The key insight is that while we're not computing gradients in traditional parameter space, we are performing gradient-based optimization in the space of scientific hypotheses, where the LLM's comparative judgments provide directional information for improvement.
>
> We hope this clarifies how our landscape differentiation approach substantively contributes to developing more efficient hypothesis optimization methods.
>
>
> **Q2**: There is no issue with using APIs. My point was rather that, given the method doesn’t introduce a new model architecture, training procedure, or dataset, the design choices relying on the API are expected to offer a high degree of novelty or utility. I’m not fully convinced that the method reaches that level yet.
>
> **Response**: Thank you for the thoughtful question, we believe that our addition of 5 baselines (including ChemCrow, MOOSE, and Reinforcement Learning) and the added expert analysis in terms of the four specific aspects (in A6) should be able to fully answer the question here. We are glad that you are satisfied with our rebuttal in A3, A6, and A7 correspondingly (e.g., the inclusion of baselines).
>
> We additionally have conducted an error analysis by experts on how the baseline is weaker than the proposed method (HHS). Two experts conduct this experiment, one checks 30 pairs of hypotheses, the other checks another 21 pairs.
>
> | Why baseline is weaker than HHS            | Count |
> |--------------------------------------------|-------|
> | Insufficient performance metrics           | 25/30 |
> | Complexity of experimental conditions      | 16/30 |
> | Insufficient explanation details           | 29/30 |
> | Inadequate description of preparation plan | 22/30 |
> | Vague research objectives                  | 28/30 |
> | Cost and scalability issues                | 13/30 |
> | Poor feasibility                           | 12/21 |
> | Errors in research plan details            | 21/21 |
> | Insufficient explanation details           | 19/21 |
> | Clear experimental system                  | 21/21 |
>
> **Q5**: The table you’ve kindly shown here looks different from Table 2 in the paper. Could you clarify where this version appears?
>
> **Response**: Yes, during the addition of baseline experiments, we discovered that the decimal precision in Table 2 was rounded to one decimal place. We have now updated the table to show two decimal places for greater precision.
>
> **Q3** & **Q4** & **Q6** & **Q7**: Thank you! Great to know that these questions are addressed!

---

> > ### Comment · Reviewer_SXMM · 2025-08-04
> > **Thank you for your response**
> >
> > Thank you for your prompt response. Here is my comment:
> >
> > Q1 & Q2
> >
> > Thank you for the clarification. I understand the details of your proposed method. However, it seems that the term “gradient” is being used here in a colloquial or broader sense. In the context of optimization, gradient is a technical term that strongly implies a specific type of computation, and using it in this way could be misleading. From what I can tell, the gradient in that technical sense is not actually being computed here.
> >
> > Regarding Section 2.3, you mention that the hierarchical method reduces computational complexity, but cases where local optima lead to global optima—as in dynamic programming—are subject to strict conditions. It does not appear guaranteed that your setting satisfies such requirements. Therefore, I interpret your use of this analogy as heuristic rather than as a theoretical justification for the method.
> >
> > As for Section 2.6, the argument appears to be more of a hypothesis than something that is directly supported or theoretically demonstrated. There is no formal guarantee of the properties you describe, and again, it’s unclear whether your specific setting strictly matches the assumptions of the analogy. Additionally, whether the hierarchical nature claimed by the authors is indeed introduced by the proposed setup remains unverified and uncertain.
> >
> > Above all, I still feel that the proposed method may not offer a significant conceptual advancement in terms of constructing or understanding new machine learning methodologies. For instance, sequential evaluation and improvement using LLMs is a relatively common idea, and using LLMs to estimate improvement direction based on before/after comparisons is also not new. In this regard, the level of conceptual novelty in your method appears to remain limited.
> >
> > ⸻
> >
> > Q3
> >
> > Regarding Q3, I feel that the inclusion of comparisons with ChemCrow and MOOSE finally brings the evaluation up to the minimally necessary level. This establishes a baseline for assessing the method’s effectiveness. That said, there are still many other methods for hypothesis generation and search, and comparisons with a broader set would further strengthen the work.
> >
> > ⸻
> >
> > Q4
> >
> > For Q4, I believe it is essential to clarify the evaluation procedure using LLMs as judges—specifically, what kind of process was employed, what exactly the judges were evaluating, and in what sense they were considered to be “satisfied.” While the suggestion that LLM-based judgments are aligned with human expectations is very positive, I would reserve judgment on its validity until those procedural details are fully explained.
> >
> > ⸻
> >
> > Q6
> >
> > Q6 has been clearly addressed. Thank you.

---

> > > ### Author Response · Authors · 2025-08-08
> > >
> > > Thank you for continuing the discussion and for your thoughtful feedback.
> > >
> > > **Q1**: It seems that the term “gradient” is being used here in a colloquial or broader sense.
> > >
> > > **A1**: Thank you for the suggestion. We agree that a term such as “gradient-like signal” is more precise, and will adopt this terminology in the revised paper to avoid ambiguity.
> > >
> > > **Q2**: Regarding Section 2.3, you mention that the hierarchical method reduces computational complexity, but cases where local optima lead to global optima—as in dynamic programming—are subject to strict conditions.
> > >
> > > **A2**: As described in Lines 133–140, the details in fine-grained scientific discovery can be organized into a set of hierarchies, where the optimal solution of a problem can be derived from the optimal solutions of its subproblems. This holds by definition if we can define a sequence of hierarchies that contain each other. (e.g. Figure 1)
> > >
> > > Even taking a step back, we do not need to frame the task exactly as dynamic programming, but we observe a structural similarity and leverage DP’s algorithmic design ideas to reduce the search complexity of fine-grained scientific discovery (line 168~169).
> > >
> > >
> > > **Q3**: As for Section 2.6, the argument appears to be more of a hypothesis than something that is directly supported or theoretically demonstrated.
> > >
> > > **A3**: Section 2.6 is correct as long as this assumption holds: “A key observation is that a hypothesis candidate’s performance at a higher hierarchy level can be viewed as an aggregated estimate—approximating an average or soft maximum—of its lower-level subspace.” (line 197~199)
> > >
> > > While this assumption is intuitively true but challenging to prove formally, we conducted additional experiments to verify it: We prompted LLMs to score hypotheses with general concepts and their corresponding specific variants on a 0–100 scale, finding an average difference of only 3.47 between the general score and the averaged specific scores (in total 204 pairs compared).
> > >
> > > It is also empirically valided by Table 1 (the *Greedy Search + Self-consistency* baseline is an ablation of HHS only without hierarchy), which directly compares "which is better local optimum found".
> > >
> > > **Q4**: What is the conceptual advancement of this paper?
> > >
> > > **A4**: Building on A1–A3, the key conceptual contribution is formulating fine-grained hypothesis discovery as an optimization problem and introducing a hierarchical search framework that (i) reduces search complexity by structuring the space into progressively refined levels, and (ii) improves solution quality by mitigating early convergence to poor local optima through level-wise smoothing of the reward landscape. This combination of problem formulation, hierarchical structuring, and analysis of ensemble-based reward landscapes goes beyond prior sequential LLM-improvement approaches.
> > >
> > > **Q5**: I feel that the inclusion of comparisons with ChemCrow and MOOSE finally brings the evaluation up to the minimally necessary level.
> > >
> > > **A5**: In addition to ChemCrow and MOOSE, we also compare with MOOSE-Chem (ICLR 2025), which is the most recent paper in this field and likely among the strongest existing methods.
> > >
> > > In direct response to your comment, we have also added SciMON (ACL 2024) as an additional baseline (soft recall: 18.57%; hard recall: 10.09%).
> > >
> > > Together with other recent frameworks, this yields **eight baselines spanning diverse paradigms**—tool-augmented, retrieval-augmented, inspiration-based, search-based, and RL-based—providing a representative benchmark for this task:
> > >
> > >
> > > **Q6**: While the suggestion that LLM-based judgments are aligned with human expectations is very positive, it is essential to clarify the evaluation procedure using LLMs as judges.
> > >
> > > **A6:** Thank you for the suggestion. We intended to include the full procedure in the main paper but were constrained by space; we will add it in the revision.
> > >
> > > **LLM-judge protocol**
> > > Each evaluation compares a pair *(ground-truth hypothesis, candidate hypothesis)*.
> > >
> > > * **Decomposition**
> > >   Both hypotheses are decomposed into methodological/experimental components using prompts co-designed with chemistry PhD students. The prompts and many decomposed examples were reviewed with them to remove duplicates and exclude non-components. Each component includes a brief description of its role, function, and context.
> > >
> > > * **Scoring**
> > >   For each ground-truth component, the LLM matches against the candidate’s components and assigns a 0–3 coverage score (0: none; 3: exact/specific match). The PhD students helped develop the prompts and audited examples to calibrate the process.
> > >
> > > * **Aggregation**
> > >   *Soft recall* is the fraction of ground-truth components with score > 0. *Hard recall* is the sum of 0–3 scores normalized by the maximum possible score.
> > >
> > > ---
> > >
> > > Overall, we sincerely appreciate your detailed feedback and constructive suggestions. We hope our responses have addressed your concerns, and we warmly welcome any further questions or comments you may have.

---

### Official Review · Reviewer_j442 · 2025-06-20

**Clarity:** 3
**Significance:** 3
**Originality:** 3
**Rating:** 4
**Confidence:** 4

**Summary:**

This paper introduces and formalizes the novel task of fine-grained scientific hypothesis discovery, which aims to generate detailed, experimentally actionable hypotheses from coarse-grained research directions. The authors frame this task as a combinatorial optimization problem over a vast, implicit space of possible experimental details. To tackle this, they propose a Hierarchical Heuristic Search (HHS) framework. This method decomposes the search problem into a series of smaller, more manageable subproblems arranged in a hierarchy of conceptual abstraction (e.g., from high-level reaction mechanisms down to specific parametric configurations in chemistry). The core intuition is that this hierarchical structure smooths the reward landscape, enabling an LLM-driven agent to more effectively navigate the search space and avoid poor local optima.

**Questions:**

Could you elaborate on the process of designing the 5-level chemistry hierarchy in Figure 1? How much expert effort was involved? More importantly, how sensitive is the performance of HHS to the specific design of this hierarchy? Can you easily extend the design to other scientific domain? For instance, what would happen if you collapsed levels 2 and 3, or used a different number of levels? This would help clarify the robustness and practical requirements of the method.

Could you provide an analysis of the computational cost (e.g., approximate number of LLM calls or total tokens processed per problem instance) for HHS compared to the baselines?

The results show HHS consistently outperforming baselines. Were there any specific types of problems or hypotheses where HHS struggled or failed to produce a superior result? An error analysis could provide valuable insights into the limitations of the current framework.

**Ethical Concerns:**

["NO or VERY MINOR ethics concerns only"]

**Final Justification:**

NA

**Paper Formatting Concerns:**

Table captions should be placed above the table.

**Quality:**

3

**Strengths And Weaknesses:**

Strengths:

The paper's primary contribution is the identification and formalization of "fine-grained scientific hypothesis discovery." This moves the field beyond generating high-level, often vague ideas towards producing concrete, testable, and actionable scientific plans.

The proposed Hierarchical Heuristic Search is well-motivated and intuitive. The analogy to coarse-to-fine refinement strategies is apt, and Figure 1 provides a concrete instantiation of this idea for the chemistry domain.

The paper is structured around three clear research questions (Q1-Q3), and the experiments are designed to answer them directly. The investigation into ensemble composition (Q3) is particularly insightful and offers practical guidance for future work.


Weaknesses:

The HHS method, with its iterative search within each level, multiple independent runs for recombination, and progression through several hierarchical levels, appears to be computationally expensive in terms of LLM API calls and tokens.

The paper title aims at scientific discovery, but the paper does not discuss how this approach would generalize to other scientific domains (e.g., biology, materials science, physics) beyond chemistry.

The "smoothing of the reward landscape" is the central theoretical claim motivating HHS. However, this claim is supported only by illustrative schematics (Figures 3 & 4) and the downstream performance of the system. While the results are strong, they are indirect evidence. The paper would be significantly strengthened by a more direct, quantitative analysis attempting to measure this smoothing effect.

---

> ### Author Rebuttal · Authors · 2025-07-31
>
> We appreciate your insightful questions and helpful suggestions. For clarity, we have organized your queries along with our responses below:
>
> ---
>
> **Q1**: The HHS method, with its iterative search within each level, multiple independent runs for recombination, and progression through several hierarchical levels, appears to be computationally expensive in terms of LLM API calls and tokens. Could you provide an analysis of the computational cost (e.g., approximate number of LLM calls or total tokens processed per problem instance) for HHS compared to the baselines?
>
> **A1**: Thank you for the question. Below is an analysis of the average reasoning steps for each method. The HHS method, while more computationally intensive, uses computation as a trade-off for improved performance. This trade-off is reasonable, as evidenced by OpenAI’s Dan Roberts, who discussed in his talk titled “9 Years to AGI? OpenAI’s Dan Roberts Reasons About Emulating Einstein” that an ideal scientific discovery system might require 8 years of inference-time computation to propose a high-quality hypothesis, emulating Einstein's process.
>
> | Method                    | Soft Recall | Hard Recall | Reasoning Steps |
> |---------------------------|-------------|-------------|-----------------|
> | Greedy                    |      16.60% |       9.92% |      9.68627451 |
> | Greedy + Self-Consistency |      31.53% |      17.73% |     67.54901961 |
> | HHS                       |      40.35% |      23.04% |     282.0392157 |
>
> **Q2**: The paper title aims at scientific discovery, but the paper does not discuss how this approach would generalize to other scientific domains (e.g., biology, materials science, physics) beyond chemistry. Could you elaborate on the process of designing the 5-level chemistry hierarchy in Figure 1? How much expert effort was involved? More importantly, how sensitive is the performance of HHS to the specific design of this hierarchy? Can you easily extend the design to other scientific domain? For instance, what would happen if you collapsed levels 2 and 3, or used a different number of levels? This would help clarify the robustness and practical requirements of the method.
>
> **A2**: Thank you for the suggestion. The method is largely domain-agnostic in its design, with the only domain-specific aspect being the hierarchy design.
> The chemistry hierarchy was designed by a chemistry PhD candidate in approximately 2 hours, based on their understanding of the chemistry research process. Similar hierarchies for other scientific disciplines can be designed by domain experts. We will discuss this further in the paper.
> To evaluate the robustness of the hierarchy design, we conducted an analysis by collapsing levels 2 and 3. The comparison results are shown below:
>
>
> | Method                   | Soft Recall | Hard Recall |
> |--------------------------|-------------|-------------|
> | HHS                      |      40.35% |      23.04% |
> | HHS (collapse hierarchy) |      37.63% |      21.79% |
>
> The results are very similar, particularly in comparison to the baselines, demonstrating the robustness of the hierarchy design.
>
>
> **Q3**: The "smoothing of the reward landscape" is the central theoretical claim motivating HHS. However, this claim is supported only by illustrative schematics (Figures 3 & 4) and the downstream performance of the system. While the results are strong, they are indirect evidence. The paper would be significantly strengthened by a more direct, quantitative analysis attempting to measure this smoothing effect.
>
> **A3**: We appreciate the reviewer's feedback on the theoretical support for our "hierarchy smooths reward" claim. We connect fine-grained hypothesis discovery to established theories in image and signal processing, where smoothing convolution (averaging pixels in an image patch) acts as a low-pass filter.
> The sufficient condition for this claim is our assumption that "LLM’s evaluation for a hypothesis with a general concept can be viewed as an aggregated estimate--approximating an average or soft maximum--of LLM’s evaluation for hypotheses with specific concepts under that general concept" (lines 197–199).
> While this assumption is commonsensically true but challenging to prove formally, we conducted additional experiments to verify it: We prompted LLMs to score hypotheses with general concepts and their corresponding specific variants on a 0–100 scale, finding an average difference of only 3.47 between the general score and the averaged specific scores (in total 204 pairs compared).
>
>
>
>
> **Q4**: The results show HHS consistently outperforming baselines. Were there any specific types of problems or hypotheses where HHS struggled or failed to produce a superior result? An error analysis could provide valuable insights into the limitations of the current framework.
>
>
> **A4**: Appendix B has four case studies including detailed expert's analysis (we will mention it in the main body of the paper). In addition to it, we conducted two more error analyses to better understand the experiment results.
>
> **Analysis 1:**
> Error analysis on the proposed method, HHS. Two chemistry PhD candidates conducted it, when one analyzed 30 hypotheses in terms of 30 research questions, the other analyzed
>  another 21 hypotheses in terms of another 21 research questions. The results are as below:
>
> | HHS Error Analysis                                     | Count |
> |--------------------------------------------------------|-------|
> | Missing key chemical substances                        | 14/30 |
> | Excessive details in characterization methods          | 28/30 |
> | Feasibility issues                                     | 18/30 |
> | Limitations of characterization methods                | 08/30 |
> | Insufficient basis for material selection              | 22/30 |
> | Lack of design comparison experiments                  | 12/30 |
> | Ignoring data validation and reproducibility           | 10/30 |
> | Severe deviation from feasibility                      |  8/21 |
> | Missing or incorrect key chemicals or reaction systems |  9/21 |
> | Incorrect explanation of chemical principles           | 12/21 |
> | Incorrect prediction of experimental system            | 10/21 |
>
> **Analysis 2:**
> Error analysis on why the baselines are weaker than HHS. Two chemistry PhD candidates conducted it: one analyzed 30 hypotheses for 30 research questions, the other analyzed 21 hypotheses for 21 research questions. The results are as follows:
>
> | Why baseline is weaker than HHS            | Count |
> |--------------------------------------------|-------|
> | Insufficient performance metrics           | 25/30 |
> | Complexity of experimental conditions      | 16/30 |
> | Insufficient explanation details           | 29/30 |
> | Inadequate description of preparation plan | 22/30 |
> | Vague research objectives                  | 28/30 |
> | Cost and scalability issues                | 13/30 |
> | Poor feasibility                           | 12/21 |
> | Errors in research plan details            | 21/21 |
> | Insufficient explanation details           | 19/21 |
> | Clear experimental system                  | 21/21 |
>
>
>
> **Q5**: Table captions should be placed above the table.
>
> **A5**: Thank you for the advice! We will adjust it in the next version.
>
> ---
> *We greatly appreciate your comprehensive feedback. We hope that our responses have satisfactorily addressed all your queries. Should you have further questions or suggestions for enhancing our manuscript, we warmly welcome your input.*

---

### Official Review · Reviewer_Rz5y · 2025-06-30

**Clarity:** 3
**Significance:** 3
**Originality:** 3
**Rating:** 4
**Confidence:** 4

**Summary:**

This paper focuses on using large language models (LLMs) to solve the problem of fine-grained scientific hypothesis discovery. Addressing the shortcomings of existing methods, which generate rough hypotheses and lack experimental details, it formalizes this task as a combinatorial optimization problem for the first time. The study proposes a hierarchical heuristic search (HHS) framework, which refines hypotheses step by step from reaction mechanisms to complete experimental configurations, and utilizes internal heuristics of LLMs for candidate editing and quality assessment. The results show that HHS outperforms significantly in LLM self-assessment, expert scoring, and recall rate of real hypotheses, and that a single strong model ensemble performs better than a diverse model ensemble.

**Questions:**

The paper mentions using the internal heuristics of LLM as a reward signal to guide hypothesis search. Then, what is the specific form of the reward function? Is it defined solely through pairwise comparisons by LLM? During the optimization process, how can the stability and consistency of the reward function be ensured? Especially in different levels of search, will there be conflicts or deviations in the reward signals? For instance, the rewards at a higher level may favor the innovativeness of concepts, while those at a lower level focus more on the feasibility of experiments. How can the differences in rewards between these levels be balanced?

The paper defines the discovery of fine-grained hypotheses as a combinatorial optimization problem, with the complexity of the search space being C{^m}_n. When m and n are large, the computational complexity will increase exponentially. Although the hierarchical search framework alleviates this problem by restricting the search space at each level, in practical applications, how to determine the optimal search range at each level? When facing very complex hypotheses (such as those involving a large number of components and parameters), can HHS effectively avoid getting stuck in local optima? Are there theoretical analyses or experimental data to support the efficiency and effectiveness of HHS in high-dimensional search spaces?

**Ethical Concerns:**

["NO or VERY MINOR ethics concerns only"]

**Final Justification:**

Thanks for the detailed reply. Most of my concerns are addressed

**Limitations:**

Yes

**Quality:**

3

**Strengths And Weaknesses:**

Strengths :
Formalizing "fine-grained scientific hypothesis discovery" as a combinatorial optimization problem, a hierarchical heuristic search (HHS) framework is proposed. It gradually refines hypotheses through a hierarchical structure (e.g., a five-level decomposition in the chemical field from reaction mechanisms to experimental configurations), effectively reducing the complexity of the search space. This hierarchical design reduces the traps of local optima by "smoothing the reward landscape", theoretically enhancing the stability of optimization.

Weaknesses:
The definition of the fine-grained assumption relies on the annotation by domain experts, which may have subjective bias, and the benchmark scale is relatively small (based on 51 chemistry papers), so the robustness needs to be verified with larger-scale data.

The HHS framework cannot guarantee finding the global optimal solution (as admitted in the manuscript), and the hierarchical division relies on domain knowledge (such as the five-level structure in chemistry), which may carry the risk of over-reliance on prior assumptions. If the domain structure is complex or difficult to stratify, the method's effectiveness may decline.

---

> ### Author Rebuttal · Authors · 2025-07-31
>
> Thank you for your insightful inquiries. In the following sections, we've structured our responses to each of your points raised.
>
> ---
>
> **Q1**: The definition of the fine-grained hypothesis relies on the annotation by domain experts, which may have subjective bias, and the benchmark scale is relatively small (based on 51 chemistry papers), so the robustness needs to be verified with larger-scale data.
>
> **A1**: We appreciate the reviewer’s concern. The hypotheses were annotated by chemistry PhD students, who are among the highest-quality annotators in the field.
>
> Scientific discovery is different from the traditional reasoning task that one discovered high-quality hypothesis could lead to significant effect. Also the data is harder to annotate. Previous efforts all have only a few dozen annotation data (purely as test data). For example, [1] [2] annotates ~50 data, [3] done by OpenAI only annotates only 20 data.
>
> [1] Large Language Models for Automated Open-domain Scientific Hypotheses Discovery, ACL 2024
>
> [2] MOOSE-Chem: Large Language Models for Rediscovering Unseen Chemistry Scientific Hypotheses, ICLR 2025
>
> [3] PaperBench: Evaluating AI's Ability to Replicate AI Research, OpenAI
>
> **Q2**: The HHS framework cannot guarantee finding the global optimal solution (as admitted in the manuscript), and the hierarchical division relies on domain knowledge (such as the five-level structure in chemistry), which may carry the risk of over-reliance on prior assumptions. If the domain structure is complex or difficult to stratify, the method's effectiveness may decline.
>
> **A2**: HHS can be seen as an optimization method. Currently none of the optimization methods is guaranteed to find the global optimum on non-convex problems.
>
> The chemistry hierarchy is designed by a chemistry phd candidate in 2 hours, and it works quite well. The hierarchy is about very general descriptions and therefore should be relatively quite robust.
>
> **Q3**: The paper mentions using the internal heuristics of LLM as a reward signal to guide hypothesis search. Then, what is the specific form of the reward function? Is it defined solely through pairwise comparisons by LLM? During the optimization process, how can the stability and consistency of the reward function be ensured? Especially in different levels of search, will there be conflicts or deviations in the reward signals? For instance, the rewards at a higher level may favor the innovativeness of concepts, while those at a lower level focus more on the feasibility of experiments. How can the differences in rewards between these levels be balanced?
>
> **A3**: The reward function is solely defined through pairwise comparisons by LLM. The pairwise comparison prompt is consistent in terms of its main body, where in different hierarchies, different specific subprompt is adopted, stressing what that hierarchy focuses on. Each hierarchy is developed with one specific subprompt.
>
> **Q4**: The paper defines the discovery of fine-grained hypotheses as a combinatorial optimization problem, with the complexity of the search space being C{^m}_n. When m and n are large, the computational complexity will increase exponentially. Although the hierarchical search framework alleviates this problem by restricting the search space at each level, in practical applications, how to determine the optimal search range at each level? When facing very complex hypotheses (such as those involving a large number of components and parameters), can HHS effectively avoid getting stuck in local optima? Are there theoretical analyses or experimental data to support the efficiency and effectiveness of HHS in high-dimensional search spaces?
>
> **A4**: In line 122 we analyze that one of the challenges for the fine-grained hypothesis discovery task is that m and n are all unknown. We deal with this challenge by adopting heuristic search in each hierarchy instead of precise search over the full space, which can avoid the need to know the full search space.
>
> The fine-grained hypothesis discovery task itself has high-dimensional search spaces, and our experiments are all conducted in it. We show that theoretically, HHS can find better local optima than the baselines, and empirically HHS does outperforms baselines.
> This work introduces the fine-grained hypothesis discovery task, and we believe that an ultimate solution for this challenge has yet to be established. We encourage the research community to explore these intriguing questions further.
>
> ---
>
> *We greatly appreciate your comprehensive feedback. We hope that our responses have satisfactorily addressed all your queries. Should you have further questions or suggestions for enhancing our manuscript, we warmly welcome your input.*

---

> > ### Comment · Reviewer_Rz5y · 2025-08-06
> >
> > Thanks for your detailed reply. I keep my score

---

> > > ### Author Response · Authors · 2025-08-08
> > >
> > > Thank you! We appreciate your consideration of our work and your feedback during the review process.

---

### Official Review · Reviewer_XFST · 2025-07-03

**Clarity:** 2
**Significance:** 3
**Originality:** 3
**Rating:** 4
**Confidence:** 3

**Summary:**

The paper investigates how LLMs can be used to generate fine-grained scientific hypotheses that could be directly implemented in laboratory settings, contrasting with prior studies that typically only produce coarse-grained ideas with low levels of detail. The authors propose a hierarchical heuristic search (HSS) approach to tackle this task, comparing it against two baseline methods that perform a flat search, namely greedy search with and without self-consistency. Using a benchmark set of 51 chemistry papers each annotated with a set of research background, a coarse-grained hypothesis, and a fine-grained hypothesis, the authors demonstrated that 1) HSS discovers superior local optima compared to the baseline methods in terms of effectiveness, novelty, detailedness and feasibility, 2) HSS-judged optimum is better aligned with the ground-truth hypotheses, and 3) using multiple instances of a strong single model is more effective at hypothesis optimization than combining different models.

**Questions:**

1. Section 2.1 - The phrase “redundant or tangential elements” is somewhat vague. Could the authors provide specific examples of these elements in the context of hypothesis generation?


2. Did the authors perform any quantitative analysis of the edits (e.g., number of insertions, deletions, or rewrites) across the hierarchy? This could strengthen their claims about the structured nature of the hierarchical heuristic search and its role in refining hypotheses.


3. Section 2.5 - How are the hierarchical levels (p) and the number of hierarchical levels defined across different research topics or domains? Moreover, how are these levels conveyed to the LLM-driven agentic process?


4. Section 3.2 - The paper mentions LLM-based dimension-specific evaluations. a) What scoring system is used for these assessments? b) How are these individual scores aggregated into an “overall” (LLM) evaluation?


5. Section 3.2 - The authors claim to observe two trade-offs. How were these conclusions drawn? Could this be better described as a correlation, rather than a causal or generalizable trade-off? Similar trade-offs could arguably exist between novelty and feasibility, especially when feasibility depends on methodological availability.


6. Section 3.3 - It is unclear whether the LLM-based evaluation of hypothesis alignment is simply matching keywords or phrases from the ground truth. If so, how do the authors ensure these matches are not out of context or semantically misleading (e.g., terms that appear similar but differ in meaning)? Concrete examples of both high and low scoring matches would help assess whether the 0–3 scale captures meaningful semantic overlap.


7. Table 3 - There appear to be a high % of ties.  It would be helpful to explicitly discuss the implications of these ties on the strength of the conclusions drawn. I would imagine that the high tie rate might suggest that differences between committees are less pronounced than the win/loss counts alone indicate.


8. Table 3 -  It has a confusing structure, as the comparison order is reversed between the left and right columns. For example, “Gemini 1.5 Flash committee vs. GPT-4o-mini committee” is in the left column, while the reverse appears in the right. Similarly, the row order for matchups is inconsistent (e.g., “Mixed committee vs. GPT-4o-mini” appears at the top on the left, but bottom on the right). Reformatting the table for consistent comparison direction and row order would improve clarity.

**Ethical Concerns:**

["NO or VERY MINOR ethics concerns only"]

**Final Justification:**

I am unable to raise my score without further access to more concrete supporting evidence and will keep my current score

**Limitations:**

The authors acknowledge that the hierarchical heuristic search (HHS) may not converge to the global optimum. However, it would strengthen the work to quantify how close HHS comes to the ground-truth hypotheses, assuming those can be treated as a proxy for the global optimum. Could the authors elaborate on whether Table 2 serves this purpose, i.e., does the recall of ground-truth components by HHS, greedy search, and self-consistent search reflect their respective distances from the global optimum?  If so, it would be helpful to explicitly frame Table 2 as a measure of this “distance to optimum” and discuss how often HHS actually recovers (or nearly recovers) the full ground-truth hypothesis. If not, are there alternative ways the authors could estimate or upper-bound the distance between generated hypotheses and the ground truth?

**Paper Formatting Concerns:**

I did not notice any major formatting issues.

**Quality:**

2

**Strengths And Weaknesses:**

Strengths:
1. The work highlights and studies a timely and important challenge on how to leverage LLMs to generate experimentally actionable scientific hypotheses, which could significantly enhance the practical utility of LLMs in scientific discovery.


2. The proposed hierarchical heuristics search approach is conceptually sound, providing a logical framework for navigating the hypothesis space effectively.


Weaknesses:
The specific questions/comments I have will be elaborated in the ‘Questions’ section but in general, the main weaknesses are:


1. Evaluation metrics lack clarity - the definitions and evaluation methodology for the evaluation metrics are insufficiently explained, which limit interpretability of the results.


2. Methodological details should provide clearer detail on how HSS is implemented.


3. The conclusions drawn from the experiments for each research question would benefit from deeper analysis and clearer justification.

---

> ### Author Rebuttal · Authors · 2025-07-31
>
> Thank you for your insightful inquiries. In the following sections, we've structured our responses to each of your points raised.
>
> ---
>
> **Q1**: Evaluation metrics & methodology can benefit from more details.
> **A1**: We thank the reviewer for this suggestion. We will add more details on evaluation metrics and methodology in the next version, as the submitted paper's dense content limited space in the main text. All relevant details, including prompts, will be updated. (Given the rebuttal space constraints, we cannot include them here.)
>
> **Q2**: The conclusions drawn from the experiments for each research question would benefit from deeper analysis.
> **A2**: Appendix B has four case studies including detailed expert's analysis (we will mention it in the main body of the paper). In addition to it, we conduct two more analyses to better understand the experiment results.
>
> **Analysis 1:**
>
> Error analysis on the proposed method, HHS. Two chemistry PhD candidates conducted it, when one analyzed 30 hypotheses in terms of 30 research questions, the other analyzed another 21 hypotheses in terms of another 21 research questions. The results are as below:
>
> | HHS Error Analysis                                     | Count |
> |--------------------------------------------------------|-------|
> | Missing key chemical substances                        | 14/30 |
> | Excessive details in characterization methods          | 28/30 |
> | Feasibility issues                                     | 18/30 |
> | Limitations of characterization methods                | 08/30 |
> | Insufficient basis for material selection              | 22/30 |
> | Lack of design comparison experiments                  | 12/30 |
> | Ignoring data validation and reproducibility           | 10/30 |
> | Severe deviation from feasibility                      |  8/21 |
> | Missing or incorrect key chemicals or reaction systems |  9/21 |
> | Incorrect explanation of chemical principles           | 12/21 |
> | Incorrect prediction of experimental system            | 10/21 |
>
> **Analysis 2:**
>
> Error analysis on why the baselines are weaker than HHS. Two chemistry PhD candidates conducted it: one analyzed 30 hypotheses for 30 research questions, the other analyzed 21 hypotheses for 21 research questions. The results are as follows:
>
> | Why baseline is weaker than HHS            | Count |
> |--------------------------------------------|-------|
> | Insufficient performance metrics           | 25/30 |
> | Complexity of experimental conditions      | 16/30 |
> | Insufficient explanation details           | 29/30 |
> | Inadequate description of preparation plan | 22/30 |
> | Vague research objectives                  | 28/30 |
> | Cost and scalability issues                | 13/30 |
> | Poor feasibility                           | 12/21 |
> | Errors in research plan details            | 21/21 |
> | Insufficient explanation details           | 19/21 |
> | Clear experimental system                  | 21/21 |
>
>
>
> **Q3**: What are “redundant or tangential elements”.
>
> **A3**: In this context, "redundant elements" refer to additional chemistry concepts that, while potentially relevant in some specific cases, are unnecessary for the hypothesis. For example, if the groundtruth hypothesis requires concepts [A, B, C, D, E], and the hypothesis includes [A, B, C, D, E, F, G, H, I], the concepts [F, G, H, I] would be considered redundant. "Tangential elements" are a subset of redundant elements, specifically referring to those that are irrelevant to the research question.
>
> **Q4**: Quantitative analysis of the edits (e.g., number of insertions, deletions, or rewrites).
>
> **A4**: We performed an additional analysis, revealing that the average number of optimization steps for HHS is 282. Of these, 113 steps involve insertions, 82 steps involve deletions, and 152 steps involve rewrites. Overall the three edits are evenly distributed across the hierarchies.
>
>
> **Q5**: How are the hierarchical levels (p) and the number of hierarchical levels defined across different research topics or domains? Moreover, how are these levels conveyed to the LLM-driven agentic process?
>
> **A5**: The chemistry hierarchy is designed by a chemistry phd candidate in 2 hours, mostly by their understanding of chemistry research. Different discipline’s hierarchy can be designed by their domain experts.
>
> These levels are explicitly defined and explained in the prompt. The code is open-sourced. Due to the space limit, please understand that we can’t list the prompt here.
>
> **Q6**: The paper mentions LLM-based dimension-specific evaluations. a) What scoring system is used for these assessments? b) How are these individual scores aggregated into an “overall” (LLM) evaluation?
>
>
> **A6**: The evaluation is based on pairwise comparisons conducted by LLMs. The "overall" evaluation has a separate prompt that includes all four dimensions, and the results for the overall evaluation are obtained independently from the individual dimension scores.
>
> **Q7**: The authors claim to observe two trade-offs. How were these conclusions drawn?
>
> **A7**: The conclusions are primarily based on the definitions of the four dimensions in the prompt. Effectiveness is supported by similar principles or methods, but this often sacrifices novelty. Feasibility (lines 255–258) is inversely related to implementation complexity, where adding details to enhance thoroughness inevitably increases complexity by introducing more elements and concepts into the hypothesis.
>
> **Q8**: It is unclear whether the LLM-based evaluation of hypothesis alignment is simply matching keywords or phrases from the ground truth. If so, how do the authors ensure these matches are not out of context or semantically misleading (e.g., terms that appear similar but differ in meaning)?
>
> **A8**: The full hypotheses are provided during the comparison to ensure complete context, and the evaluation prompt explicitly emphasizes this requirement. Each generated hypothesis is approximately 600 words and requires domain-specific chemistry knowledge for proper understanding. While providing specific examples here may not be feasible, we extensively consulted with chemistry experts during the development of the evaluation framework to ensure its quality. A significant portion of our research efforts was dedicated to developing a robust evaluation framework.
>
> **Q9**: Table 3 - There appears to be a high % of ties. It would be helpful to explicitly discuss the implications of these ties on the strength of the conclusions drawn. I would imagine that the high tie rate might suggest that differences between committees are less pronounced than the win/loss counts alone indicate.
>
> **A9**: Thank you for your insightful observation. The conclusion focuses on determining which committee is better, and the tie rate does not affect the qualitative outcome. The conclusion is supported by multiple pairwise committee comparisons, many of which show clear win/loss discrepancies. We include the tie results to provide a more comprehensive understanding of the comparison process.
>
> **Q10**: Table 3 - It has a confusing structure, as the comparison order is reversed between the left and right columns. Reformatting the table for consistent comparison direction and row order would improve clarity.
>
> **A10**: Thank you for your careful review and helpful suggestion. We will reformat Table 3 for improved clarity and consistency.
>
>
> **Q11**: Could the authors elaborate on whether Table 2 serves this purpose, i.e., does the recall of ground-truth components by HHS, greedy search, and self-consistent search reflect their respective distances from the global optimum? If so, it would be helpful to explicitly frame Table 2 as a measure of this “distance to optimum” and discuss how often HHS actually recovers (or nearly recovers) the full ground-truth hypothesis.
>
> **A11**: Thank you for the insightful suggestion. Yes, the recall of ground-truth components reflects the respective distances from the global optimum. We will explicitly frame Table 2 as a measure of "distance to optimum" and discuss how often HHS recovers (or nearly recovers) the full ground-truth hypothesis in the paper.
>
>
> ---
> *We greatly appreciate your comprehensive feedback. We hope that our responses have satisfactorily addressed all your queries. Should you have further questions or suggestions for enhancing our manuscript, we warmly welcome your input.*

---

### Official Review · Reviewer_dwWU · 2025-07-03

**Clarity:** 2
**Significance:** 2
**Originality:** 1
**Rating:** 3
**Confidence:** 3

**Summary:**

This paper introduces the task of fine-grained scientific hypothesis discovery and proposes a Hierarchical Heuristic Search (HHS) framework that leverages LLMs both to edit and evaluate hypotheses across multiple abstraction levels. Experiments on a small chemistry benchmark show that HHS outperforms greedy baselines in LLM self-assessments and expert rankings.

**Questions:**

No

**Ethical Concerns:**

["NO or VERY MINOR ethics concerns only"]

**Final Justification:**

Most of my questions have been addressed. However, I still find the paper difficult to follow due to the limited literature review. This lack of contextualization makes it challenging to compare the proposed approach with existing work and to clearly identify the paper’s contributions. Expanding the related work section and clarifying how this work differs from or builds upon prior studies would significantly improve the paper's clarity and accessibility.

**Limitations:**

No. Please do a thorough literature review and compare with SOTA methods.

**Quality:**

1

**Strengths And Weaknesses:**

### Strengths

1. Clear task definition & dataset – The authors articulate why coarse hypotheses are insufficient and release a contamination-controlled benchmark with expert annotations, filling a recognised gap in the literature.

2. Coherent agentic framework – HHS is a self-contained, reproducible pipeline that factors an intractable search into tractable sub-problems and provides intuitive smoothing analysis.

### Weaknesses
1. The literature review is exceptionally thin for a NeurIPS-level submission—only ~15 references are listed, many of them generic model or dataset announcements rather than directly relevant prior art. Key recent work on hierarchical planning, tree-of-thought search, and agent-based scientific discovery published in NeurIPS/ICLR/ACL 2024-25 is absent. This shallow bibliography suggests the authors have not situated their contribution within the full landscape of related methods and undermines claims of novelty and significance.

2. The contribution’s novelty is overstated. Prior systems, such as Chemcrow [1] and other retrieval-plus-refine agents already tackle hypothesis generation (some explicitly optimise for novelty and detail); the paper does not cite or benchmark against them nor clarify conceptual differences, leaving incremental value uncertain.

3. HHS is essentially a coarse-to-fine or evolutionary search variant; the “hierarchy smooths reward” claim is supported only by schematic plots, lacking formal metrics or convergence theory, so theoretical depth is thin.

4. Baselines omit standard combinatorial optimisers (reinforcement learning, MCTS, genetic algorithms) and recent retrieval-augment-edit pipelines, so HHS’s superiority is unproven outside the chosen comparisons.

[1] Bran, Andres M., et al. "Chemcrow: Augmenting large-language models with chemistry tools." arXiv preprint arXiv:2304.05376 (2023).

---

> ### Author Rebuttal · Authors · 2025-07-31
>
> We appreciate your insightful questions and helpful suggestions. For clarity, we have organized your queries along with our responses below:
>
> ---
>
> **Q1**: Only ~15 references are listed, Key recent work on agent-based scientific discovery published in NeurIPS/ICLR/ACL 2024-25 is absent.
>
> **A1**: We thank the reviewer for raising concerns about the references. As this work introduces the novel task of fine-grained scientific hypothesis discovery—for which no prior research exists—there are inherently fewer directly relevant papers to cite.
>
> We did cite key recent works on agent-based scientific discovery published in NeurIPS/ICLR/ACL 2024-25, including:
>
> [1] Scimon: Scientific inspiration machines optimized for novelty, ACL 2024.
> [2] Large Language Models for Automated Open-domain Scientific Hypotheses Discovery, ACL 2024.
> [3] Large language models are zero shot hypothesis proposers, COLM 2024.
> [4] MOOSE-Chem: Large Language Models for Rediscovering Unseen Chemistry Scientific Hypotheses, ICLR 2025.
> [5] Can llms generate novel research ideas? A large-scale human study with 100+ NLP researchers, ICLR 2025.
> [6] NOVA: An Iterative Planning Framework for Enhancing Scientific Innovation with Large Language Models, ACL 2025.
>
>
> **Q2**: Previous work like Chemcrow already works on hypothesis generation.
>
> **A2**: We thank the reviewer for highlighting ChemCrow. While previous works like ChemCrow address hypothesis generation, it does not focus on fine-grained hypothesis discovery. We did not adopt it as a baseline initially because they won’t have a comparable performance on providing the hypothesis details. We have **added 5 additional baselines** including ChemCrow [7] as well as four other strong baselines (three published in top conferences in the recent two years, and one combinatorial optimization optimizer as required in Q3), with newly added baselines boldened. We will refer to them in the paper.
>
>
> |                            | Soft Recall | Hard Recall |
> |----------------------------|-------------|-------------|
> | **ChemCrow[7]**               |      12.28% |       7.20% |
> | **Reinforcement Learning** |      16.76% |      10.28% |
> | **[3]**                     |      19.57% |      11.15% |
> | **MOOSE[2]**                  |      20.04% |      11.76% |
> | **MOOSE-Chem[4]**             |      19.99% |      11.98% |
> | Greedy                     |      16.60% |       9.92% |
> | Greedy + Self-Consistency                         |      31.53% |      17.73% |
> | HHS                         |      **40.35%** |     **23.04%** |
>
> [7] ChemCrow: Augmenting large-language models with chemistry tools, Nature
>
>
>
> **Q3**: Baselines omit standard combinatorial optimisers such as reinforcement learning, MCTS, genetic algorithms.
>
> **A3**: We thank the reviewer for suggesting additional combinatorial optimizers as baselines. As fine-grained scientific hypothesis discovery is a new task introduced in this paper, and while we formalize it as a combinatorial optimization problem, the search space is inherently unknown (line 122: "n is unknown"), unlike traditional problems.
>
> MCTS and genetic algorithms require a clearly defined search space, precluding direct implementation here. We are aware that prior works like [8][9] use MCTS, but in tasks with explicit search spaces.
>
> Reinforcement learning, however, does not require a defined search space, and we have implemented it as shown in the Table of A2.
>
>
> [8] Monte Carlo Thought Search: Large Language Model Querying for Complex Scientific Reasoning in Catalyst Design, EMNLP 2023.
> [9] CHEMREASONER: Heuristic Search over a Large Language Model's Knowledge Space using Quantum-Chemical Feedback, ICML 2024.
>
>
>
>
> **Q4**: HHS is essentially a coarse-to-fine or evolutionary search variant.
>
> **A4**: We appreciate the reviewer's observation that HHS incorporates coarse-to-fine and evolutionary elements. However, our core contributions lie in formalizing the novel fine-grained hypothesis discovery task as a combinatorial optimization problem and developing HHS to optimize it effectively for scientific discovery. No prior LLM-based scientific discovery work has framed the task through an optimization lens. Moreover, HHS advances coarse-to-fine search by adapting it to challenging, language-based open-ended tasks like hypothesis generation, where such implementations were previously unclear.
>
> **Q5**: The “hierarchy smooths reward” claim is supported only by schematic plots, lacking formal metrics or convergence theory.
>
> **A5**:
> We appreciate the reviewer's feedback on the theoretical support for our "hierarchy smooths reward" claim. We connect fine-grained hypothesis discovery to established theories in image and signal processing, where smoothing convolution (averaging pixels in an image patch) acts as a low-pass filter.
>
> The sufficient condition for this claim is our assumption that "LLM’s evaluation for a hypothesis with a general concept can be viewed as an aggregated estimate--approximating an average or soft maximum--of LLM’s evaluation for hypotheses with specific concepts under that general concept" (lines 197–199).
>
> While this assumption is commonsensically true but challenging to prove formally, we conducted additional experiments to verify it: We prompted LLMs to score hypotheses with general concepts and their corresponding specific variants on a 0–100 scale, finding an average difference of only 3.47 between the general score and the averaged specific scores (in total 204 pairs compared).
>
> ---
> *We greatly appreciate your comprehensive feedback. We hope that our responses have satisfactorily addressed all your queries. Should you have further questions or suggestions for enhancing our manuscript, we warmly welcome your input.*

---

> > ### Comment · Reviewer_dwWU · 2025-08-05
> >
> > Thank you to the authors for the detailed response. Most of my concerns have been addressed, and I will accordingly raise my score. However, I would strongly encourage the authors to conduct a more thorough literature review, including related works and similar techniques that are relevant to this study. The current version may be difficult to follow for readers who are not already familiar with this specific area, and it remains unclear what the precise contributions of the paper are in relation to existing work.

---

> > > ### Author Response · Authors · 2025-08-08
> > >
> > > Great to know that most of your concerns have been addressed—and thank you for raising the rating.
> > >
> > > Regarding your remaining suggestion on the literature review: while our work proposes a new task and, to our knowledge, no prior studies address it directly, we agree that including more related works and similar techniques will help readers unfamiliar with the area better follow the paper and situate our contributions. We will expand the literature review in the revision to cite and discuss additional relevant works. Thank you again for your thoughtful feedback.

---

### Author Response · Authors · 2025-08-09
**Thank All Reviewers**

To all reviewers:

We would like to thank all reviewers for their thoughtful insights and valuable comments.

We summarize the contribution of this paper below:

1. We introduce and formalize fine-grained scientific hypothesis discovery as a combinatorial optimization problem, and release a post-2024 chemistry benchmark with expert-annotated fine-grained hypotheses, explicitly designed to prevent data contamination for current LLMs.

2. We systematically investigate this task through three foundational research questions: (Q1) how to best leverage an LLM’s internal heuristics for fine-grained hypothesis generation; (Q2) whether LLM-preferred hypotheses align more closely with ground-truth expert hypotheses; and (Q3) whether ensembles of diverse LLMs provide a better reward landscape compared to repeated use of the strongest single model.

3. We propose a hierarchical search method over levels of conceptual abstraction, which smooths the reward landscape and reduces search complexity at each hierarchy level. Empirically, it consistently outperforms strong baselines in both LLM self-evaluation, expert evaluation, and recall against annotated ground-truth hypotheses.

We are excited that you recognized our contributions. We quote correspondingly as below:

1. “Clear task definition & dataset” [reviewer dwWU]; “The task studied could significantly enhance the practical utility of LLMs in scientific discovery” [reviewer XFST]; “It formalizes this task as a combinatorial optimization problem for the first time.” [reviewer Rz5y]; “The paper's primary contribution is the identification and formalization of ‘fine-grained scientific hypothesis discovery.’ This moves the field beyond generating high-level, often vague ideas towards producing concrete, testable, and actionable scientific plans.” [reviewer j442]; “The authors’ attempt to address the gap between such abstract hypotheses and their concrete implementation is a commendable aspect of the work.” [reviewer SXMM]

2. “The paper is structured around three clear research questions (Q1-Q3), and the experiments are designed to answer them directly. The investigation into ensemble composition (Q3) is particularly insightful and offers practical guidance for future work.” [reviewer j442]

3. “HHS is a self-contained, reproducible pipeline that factors an intractable search into tractable sub-problems and provides intuitive smoothing analysis” [reviewer dwWU]; “ HHS is conceptually sound, providing a logical framework for navigating the hypothesis space effectively” [reviewer XFST]; “HHS… effectively reducing the complexity of the search space … reduces the traps of local optima … theoretically enhancing the stability of optimization” [reviewer Rz5y]; “The proposed Hierarchical Heuristic Search is well-motivated and intuitive. The analogy to coarse-to-fine refinement strategies is apt, and Figure 1 provides a concrete instantiation of this idea for the chemistry domain.” [reviewer j442]

We are grateful that you also found that:

1. “Their approach of interpreting LLM-based generation as a process akin to search and building their methodology around this analogy is both novel and intellectually stimulating.” [reviewer j442].
2. “incorporating human evaluations, which strengthens the credibility of their findings.” [reviewer j442].
3. “their focus on evaluating hypotheses in terms of their ‘detailedness’ is an insightful and original perspective that adds value to the study.” [reviewer j442].
4. “this is an excellent piece of work that contributes meaningfully to the field of AI-driven hypothesis generation” [reviewer j442]

We also appreciate many helpful suggestions, based on which we have improved our manuscript. The main added experiments & analyses are:

1. Addition of 6 strong baselines: 5 of them are published on top conferences in recent 2 years, 1 of them is a classic combinatorial optimization method. All of them have huge performance gap with the proposed method.

2. Error analysis of where the baselines perform worse than HHS performed by two chemistry phd students.

3. Error analysis of where HHS can be further improved performed by two chemistry phd students.

4. Empirical analysis and support to the assumption “hierarchy design smoothens optimization landscape”.

5. Addition of expert evaluation of Table 1 in terms of each of the specific aspect: effectiveness, novelty, feasibility, and detailedness.

6. Analysis of the number of reasoning steps leveraged by HHS and the baselines.

7. Analysis of the effect of hierarchy variant to the performance of HHS.

We will also quote more previous papers, and add more details of the method and evaluation to the paper.

We would again like to thank all 5 reviewers for their time and effort. We are grateful for the in-depth discussions, and we have work hard to reply to every message. We hope that our changes adequately address all concerns. We are open to new discussions whenever possible.

Sincerely,
Authors

---

### Note · Authors · 2025-08-13

Dear AC and Reviewers,

We sincerely thank all reviewers and the AC for their thoughtful engagement with our submission. The contributions of the paper are outlined in the “Thank All Reviewers” comment, and we are pleased to note that each contribution has been endorsed by at least one reviewer, with the relevant comments from the reviewers carefully selected and included in the comment.

From the authors' perspective, we believe we have thoroughly addressed all the concerns raised by the reviewers. We have responded to each question, incorporated numerous additional experiments to reinforce our arguments, and engaged in detailed discussions with the reviewers who provided feedback. Due to word limitations, we were unable to include all responses here, but we have outlined the key added experiments and analyses in the "Thank All Reviewers" comment. Notably, we introduced six additional strong baselines in the rebuttal, none of which come close to matching the performance of our proposed method. Thanks to the reviewers' constructive feedback, and with the new experiments conducted during the rebuttal, every claim in the paper is now well supported by at least one experiment.

We firmly believe that our work makes a meaningful contribution to the community. We are also deeply grateful for the kind words from the reviewers, some of which we quote below: “filling a recognized gap in the literature” [reviewer dwWU], “significantly enhances the practical utility of LLMs in scientific discovery” [reviewer XFST], “formalizes this task as … for the first time” [reviewer Rz5y], and “moves the field beyond … towards …” [reviewer j442].

Sincerely,
Authors

---

### Decision · Program_Chairs · 2025-09-17

**Decision:**

Accept (poster)

**Comment:**

This paper introduces a methodology for using LLMs to propose and refine scientific hypotheses in a hierarchical manner, across multiple layers of abstraction. The paper articulates the problem clearly and the motivation and description of the method is clear. The general idea of iterative LLM-driven coarse-to-fine refinement in a hypothesis makes sense and the authors investigate the idea on a limited but interesting chemistry problem. The results are convincingly superior to a number of baselines. During the author/reviewer discussion, the overall technique was clarified and in particular new baselines were introduced that substantially improved the experimental soundness of the paper.